# GLANCE FOR CONTEXT: LEARNING WHEN TO LEVERAGE LLMS FOR NODE-AWARE GNN-LLM FUSION

**Donald Loveland, Yao-An Yang & Danai Koutra**
Department of Computer Science and Engineering
University of Michigan
Ann Arbor, MI, USA
`{dlovelan, ayayang, dkoutra}@umich.edu`

## ABSTRACT

Learning on text-attributed graphs has motivated the use of Large Language Models (LLMs) for graph learning. However, most fusion strategies are applied uniformly across all nodes and attain only small overall performance gains. We argue this result stems from aggregate metrics that obscure *when LLMs provide benefit*, inhibiting actionable signals for new strategies. In this work, we reframe LLM–GNN fusion around nodes where GNNs typically falter. We first show that performance can significantly differ between GNNs and LLMs, with each excelling on distinct structural patterns, such as local homophily. To leverage this finding, we propose **GLANCE** (**G**NN with **LLM** **A**ssistance for **N**eighbor- and **C**ontext-aware **E**mbeddings), a framework that invokes an LLM to refine a GNN's prediction. GLANCE employs a lightweight router that, given inexpensive per-node signals, decides whether to query the LLM. Since the LLM calls are non-differentiable, the router is trained with an advantage-based objective that compares the utility of querying the LLM against relying solely on the GNN. Across multiple benchmarks, GLANCE achieves the best performance balance across node subgroups, achieving significant gains on heterophilous nodes (up to $+13\%$) while simultaneously achieving top overall performance. Our findings highlight the value of adaptive, node-aware GNN-LLM architectures, where selectively invoking the LLM enables scalable deployment on large graphs without incurring high computational costs.

## 1 INTRODUCTION

Text is rarely consumed in isolation: scientific papers cite related work (Ciotti et al., 2015; Mccallum et al., 2000; Cohan et al., 2018), users browse descriptions of co-purchased e-commerce items (Jin et al., 2024; McAuley & Yang, 2016; Ezeife & Karlapalepu, 2023), and social media posts reply to one another (Wu et al., 2017; Yang & Leskovec, 2011). These interactions form text-attributed graphs (TAGs), where nodes represent text and edges capture relationships, enabling joint reasoning over content and structure (Yang et al., 2015; Zhao et al., 2023). Historically, TAGs have been processed by feeding shallow text features, such as TF–IDF or static word embeddings (Stephen et al., 2022; Mikolov et al., 2013), into Graph Neural Networks (GNNs). Recently, the success of Large Language Models (LLMs) has motivated hybrid architectures that leverage LLMs for graph learning, combining LLMs' semantic reasoning with GNNs' structural learning (Chen et al., 2024a; Wang et al., 2025). However, the majority of hybrid systems typically apply a single fusion strategy across all nodes in a graph, overlooking per-node variations in semantic quality and structural attributes (Wu et al., 2024). This uniform application of LLMs across a graph can waste expensive LLM calls on nodes already well modeled by the GNN, producing poor accuracy-efficiency tradeoffs (Liu et al., 2025).

A key weakness of uniform fusion is its failure to leverage the complementary strengths of GNNs and LLMs. GNNs tend to perform well when nodes have a high degree and their neighbors exhibit homophily (i.e., connected nodes share labels). (McPherson et al., 2001; Yan et al., 2022). Yet, these properties do not typically hold in real-world TAGs (Ma et al., 2020; Loveland et al., 2024; Zhou et al., 2020). While advanced GNN designs have attempted to address these concerns (Zhu et al., 2020; Abu-El-Haija et al., 2019; Veličković et al., 2018), recent evidence shows they remain insufficient for handling these challenging structures (Loveland & Koutra, 2025; Mao et al., 2023; Du et al., 2022). On the other hand, LLMs exhibit strong generalization in low-shot settings, making them well suited for the challenging nodes that degrade GNNs (Chen et al., 2024a;

Peng et al., 2024). However, leveraging LLMs for graphs often leads to the distortion of structural relationships due to serializing the graph into text (Liu et al., 2023; Firooz et al., 2025; Wang et al., 2025). This limitation is particularly detrimental for graphs governed by simpler structural signals where the LLM may add unnecessary complexity and even degrade performance relative to GNNs (Liu et al., 2025; Wang et al., 2025).

While incremental performance gains with high computational costs make LLMs seem ill-suited for graph learning, we argue this stems not from inherent deficiencies, but how LLMs are used. Our hypothesis is that gains in regions difficult for GNNs are offset by losses elsewhere. Thus, we focus on utilizing LLMs only where GNNs struggle, prioritizing structurally difficult nodes whose errors are often masked in aggregate metrics. This emphasis promotes equity across the graph, countering harmful GNN inductive biases (Wang et al., 2022; Agarwal et al., 2021). This leads us to our core research question: ***How***, and for *which* **nodes, should we leverage LLMs to complement and bolster GNNs?**

To answer this question, we first identify signals that inform when GNNs vs. LLMs succeed. Since LLM routing has only recently been studied, we systematically assess existing methods (Qiao et al., 2025; Jaiswal et al., 2024; Chen et al., 2024b), finding highly variable performance, sometimes worse than random routing. Through a node-level analysis on local homophily and relative degree, two properties known to impact GNNs (Yan et al., 2022; Subramonian et al., 2024), we find GNNs excel in high-homophily, well-connected regions, while LLMs dominate as homophily or degree decreases. Notably, local homophily emerges as a strong predictor of model advantage, offering a principled prior for routing.

Motivated by this, we propose **GLANCE** (**G**NN with **LLM A**ssistance for **N**eighbor- and **C**ontext-aware **E**mbeddings), a fusion strategy that preserves GNN efficiency on easy nodes while selectively leveraging LLMs on hard ones. GLANCE first encodes nodes with a pre-trained GNN, then passes lightweight routing features to a cost-aware policy that decides whether to query the LLM – figuratively "glancing" at the LLM for additional context. For routed nodes, LLM embeddings are fused with GNN embeddings via a small refiner head. To train the non-differentiable router, we introduce an advantage-based strategy that rewards beneficial queries to the LLM. We find that GLANCE is able to consistently outperform previous benchmarks, producing more robust predictions across the spectrum of homophily levels. This ability to achieve higher performance while reducing LLM usage, and thus computational cost, highlights the value of node-aware GNN-LLM fusion when scaling to large, diverse TAGs. Our contributions are below:

- **Current Limitations: *Which* Nodes to Route.** We provide a systematic look at LLM routing and show that current heuristics are brittle. By stratifying nodes by different properties, we reveal the complementary performance of GNNs and LLMs over local homophily and identify for which nodes LLMs are beneficial.
- **New GNN-LLM Method: *How* to Select Nodes.** We introduce GLANCE, a cost-aware framework that learns when to query the LLM, minimizing unnecessary LLM calls and preserving scalability.
- **Comprehensive Empirical Analysis.** Across four diverse TAG datasets, GLANCE consistently outperforms state-of-the-art GNNs and GNN-LLM hybrids, yielding robust predictions across homophily levels with gains of up to $+13.0\%$ on heterophilous pockets and overall performance gains of up to $+0.9\%$.

## 2    RELATED WORK

We briefly discuss the most relevant work here, and provide more details in Appendix A.

**Static GNN-LLM Fusion.**    Prior work largely follows two paradigms: LLMs-as-Enhancers and LLMs-as-Predictors (Chen et al., 2024a; Li et al., 2024). Enhancer methods enrich node features with text embeddings or external knowledge (Wu et al., 2024; He et al., 2024), while predictor methods replace the GNN with the LLM, directly classifying serialized graph inputs (Wu et al., 2024; Wang et al., 2025). LLM-as-Predictor methods can overcome GNN biases, but struggle with topology encoding and prompt length (Firooz et al., 2025). Other fusion strategies draw from Mixture-of-Experts (MoE) (Cai et al., 2025; Wang et al., 2023), but have not explored mixing GNNs and LLMs. Recent work has considered LLMs to route among GNNs (Jiang & Luo, 2025), yet this still confines performance to GNN biases. In contrast, our approach adopts a selective paradigm, invoking the LLM where it is expected to help, combining the strengths of both models.

**Adaptive GNN-LLM Fusion.**    A challenge for LLM–GNN models is inference cost, especially when LLMs are used in the forward pass for all nodes (Wu et al., 2024). While cost reduction via sampling or distillation helps (Fang et al., 2023; Chen et al., 2024a), scalability remains limited by universal LLM calls. Only a few recent works explore adaptive GNN-LLM fusion, where the LLM is invoked selectively. E-LLaGNN

routes nodes with fixed heuristics (e.g., degree, centrality) (Jaiswal et al., 2024), but requires manual tuning. LOGIN uses GNN uncertainty to rewire difficult nodes (Qiao et al., 2025), though this can erase useful heterophilous links (Luan et al., 2021). LLM-GNN uses clustering density as a proxy for difficulty (Chen et al., 2024b). Our approach differs by: (1) systematically identifying structural properties predictive of LLM benefit, learning to route without preset thresholds; (2) preserving graph structure rather than editing it; and (3) directly training a router under non-differentiable queries, rather than altering data to create influence.

## 3 PRELIMINARIES

A TAG is defined as $\mathcal{G} = (\mathcal{V}, \mathcal{E}, T, Y)$, where $\mathcal{V}$ is the set of $n = |\mathcal{V}|$ nodes, $\mathcal{E} \subseteq \mathcal{V} \times \mathcal{V}$ is the set of edges, $T = \{t_v\}_{v \in \mathcal{V}}$ is the text associated with each node $v$, and $Y = \{y_v\}_{v \in \mathcal{V}}$ is the set of node labels. The task is to learn a model $\psi : (\mathcal{G}, T) \to Y$ that predicts node labels $y_v$ from both structure and text. Next we outline the strategies to parameterize and learn $\psi$.

**Graph Neural Networks.** Node representations are learned by aggregating messages from neighbors by:

$$\mathbf{h}_v^{(\ell)} = \text{UPDATE}^{(\ell)}\left(\mathbf{h}_v^{(\ell-1)}, \text{AGGREGATE}^{(\ell)}\left(\{\mathbf{h}_u^{(\ell-1)} : u \in \mathcal{N}(v)\}\right)\right), \quad \hat{y}_v = \arg\max \text{MLP}(\mathbf{h}_v^{(L)}).$$

for a node $v$ at layer $\ell$. Additionally, $\mathbf{h_v}^{(0)} = \mathbf{x}_v$ (a feature vector from $t_v$), $\mathcal{N}(v)$ denotes the neighbors of node $v$, and AGGREGATE and UPDATE define the GNN's operations. In practice, AGGREGATE is a permutation-invariant function (e.g. sum or mean) and UPDATE is a multi-layer perceptron (MLP). After $L$ layers, the prediction is derived from an MLP head. Despite their success, GNNs can struggle on certain structural patterns, such as low degree and heterophily (Loveland et al., 2024; Mao et al., 2023; Yan et al., 2022), motivating architectures with higher-order aggregation, residual connections, and adaptive message passing (Chen et al., 2020; Zhu et al., 2020). We leverage GNNs that adopt these designs later in our analysis.

**Large Language Models.** Let LLM$(\cdot)$ denote a pre-trained language model. Given a node $v$ with text $t_v$, and optionally neighbor attributes $\{t_u : u \in \mathcal{N}(v)\}$, the LLM can either (i) generate an embedding $\mathbf{z}_v = \text{LLM}_{\text{embed}}(p_v)$ from a prompt $p_v$, optionally including neighborhood context, or (ii) directly predict the label $\hat{y}_v = \text{LLM}_{\text{predict}}(p_v)$. Embedding LLMs allow the output $z_v$ to be easily combined with a GNN's output or other downstream classifiers, but require a prediction head. In contrast, direct prediction avoids this head but risks hallucinated labels and often still needs fine-tuning. In this work, we adopt the embedding setting, which supports seamless integration with GNN representations.

## 4 WHICH NODES TO ROUTE: CURRENT LIMITATIONS & OPPORTUNITIES

We begin by examining *which nodes* should be routed, reviewing prior routing strategies and highlighting their limitations. Then, building on insights from the GNN literature, particularly the challenges with GNNs applied to low-degree nodes and heterophilous neighborhoods (Yan et al., 2022; Tang et al., 2020), we show that LLMs can complement GNNs under these conditions. Ultimately, we find that homophily emerges as a strong indicator for LLM benefit, offering a promising opportunity to improve graph learning.

### 4.1 LIMITATIONS OF CURRENT ROUTING STRATEGIES

Based on previous work, we analyze strategies to route nodes to an LLM. We consider node degree $d_v$ (Jaiswal et al., 2024), clustering density (C-density) (Chen et al., 2024b), and uncertainty from dropout (Qiao et al., 2025). For each strategy, we route the top-$k\%$ of nodes to a fine-tuned LLM under three criteria: (i) low degree, (ii) low density, and (iii) high uncertainty.

**Experimental Setup.** We evaluate on Cora (Mccallum et al., 2000), Pubmed (Sen et al., 2008a), and Arxiv23 (He et al., 2024) with processing details in Section D. We train two backbones, GCN and GCNII, using the dataset's original features (denoted as "Std.") and enhanced features (denoted as "Enh.") generated via Qwen3-8B. GCN serves as a traditional baseline, while GCNII introduces new designs to better handle over-smoothing and other challenging graph properties. Together, they provide a contrast between a simple message-passing framework and a stronger state-of-the-art backbone. For LLM-as-predictor, we fine-tune Qwen3-8B with an MLP head for node classification. LLM prompting and training for both LLM-as-Enhancer and LLM-as-Predictor are given in Section B. Training details and hyperparameters for the LLMs and GNNs are provided in Section C. For the routing strategies in Table 1, we freeze the GNN and LLMs and route using the metric.

We assess heuristics with a **net correction score** (NCS), capturing the benefit of routing. For a routed set $R$, let $WC$ be the nodes *wrong* under the GNN but became *correct* after routing to the LLM, and $CW$ be the set of nodes were *correct* but became *wrong*. We define $\text{NCS} = (|WC| - |CW|)/|R|$ where $\text{NCS} = 1$ means the LLM fixes all routed nodes and $-1$ means the LLM harmed every routed node.

**Findings.** Table 1 highlights the difficulty to identify a single routing signal that generalizes across datasets. On Pubmed and Arxiv23, uncertainty appears promising, achieving the best NCS in every setting. Yet on Cora, the same strategy consistently produces negative NCS, often performing worse than a random router. Other heuristics such as degree and C-density similarly fluctuate in effectiveness, with no strategy performing robustly across all datasets. Together, these results underscore the **limitations of static heuristics**: while they can succeed in isolated cases, **their performance is dataset-dependent**. This motivates the need for a more principled and transferable routing signal, rather than manually chosen rules.

**Table 1:** NCS for C-density, $d_v$, and uncertainty, as compared to a random router, higher is better. The number of routed nodes is in parenthesis. The highest NCS score is **bold** for each setting.

|  |  | Cora | | | Pubmed | | | Arxiv23 | | |
|---|---|---|---|---|---|---|---|---|---|---|
|  |  | **10%** | **15%** | **20%** | **10%** | **15%** | **20%** | **10%** | **15%** | **20%** |
|  | *Routing Strat.* | (68) | (102) | (136) | (493) | (740) | (986) | (489) | (734) | (978) |
| **GCN Enh.** | Random | **-0.02** | -0.06 | -0.04 | 0.06 | 0.05 | 0.04 | 0.05 | 0.04 | 0.04 |
|  | C-density | **-0.02** | -0.03 | -0.03 | 0.05 | 0.06 | 0.05 | 0.07 | 0.07 | 0.06 |
|  | Degree | -0.04 | **-0.01** | -0.02 | 0.04 | 0.04 | 0.04 | 0.04 | 0.05 | 0.05 |
|  | Uncertainty | -0.09 | -0.03 | **-0.01** | **0.20** | **0.18** | **0.17** | **0.15** | **0.13** | **0.13** |
| **GCNII Enh.** | Random | **0.00** | **0.01** | **-0.01** | 0.01 | 0.01 | 0.01 | -0.01 | 0.00 | 0.00 |
|  | C-density | **0.00** | -0.03 | -0.03 | 0.02 | 0.02 | 0.02 | 0.03 | 0.03 | 0.03 |
|  | Degree | -0.03 | -0.03 | -0.02 | 0.03 | 0.03 | 0.03 | -0.03 | -0.02 | -0.01 |
|  | Uncertainty | -0.04 | -0.02 | -0.03 | **0.09** | **0.08** | **0.08** | **0.05** | **0.04** | **0.05** |

## 4.2 STRUCTURAL OPPORTUNITIES FOR ROUTING

Given that heuristic quality can be dataset-dependent, we aim to find a routing signal that aligns with the strengths of GNNs and LLMs. We first analyze metrics known to degrade GNN performance and then utilize these to route, showing that homophily is a strong indicator for when LLMs improve performance.

### 4.2.1 ON THE COMPLEMENTARY CAPABILITIES OF LLMS AND GNNS

To characterize the unique benefits of LLMs and GNNs, we take the models from Table 1 and perform a stratified analysis using *relative degree* $\bar{d}_v$ and *local homophily* $h_v$. When $\bar{d}_v > 1$, $v$ tends to have higher degree than its neighbors, and lower degree otherwise. Low $h_v$ indicates $v$ is heterophilous. Mathematically,

$$\bar{d}_v = \frac{1}{|\mathcal{N}(v)|} \sum_{u \in \mathcal{N}(v)} \sqrt{\frac{d_v + 1}{d_u + 1}}, \qquad h_v = \frac{1}{|\mathcal{N}(v)|} \sum_{u \in \mathcal{N}(v)} \mathbf{1}[y_u = y_v].$$

Both properties are known to influence GNNs (Yan et al., 2022; Subramonian et al., 2024), but have received less attention for LLM-graph reasoning. We compare models across these properties to understand how each paradigm responds.

**Findings.** In Figure 1, we study performance for groups of nodes stratified by $h_v$ and $\bar{d}_v$. We find that **LLM models tend to produce significantly higher performance over heterophilous and low-degree nodes**, e.g., achieving up to a $20.4\%$ performance increase compared to the next best model, GCNII with LLM enhancement on Cora. Additional results given for Pubmed in Section E.4 with similar trends. In Figure 4 (Section E.5), we further find that homophily and degree can interplay with one another, where performance differences can reach upwards of $30.1\%$ between subpopulations stratified by both degree and local homophily. With these findings, we next study if homophily can be useful for routing.

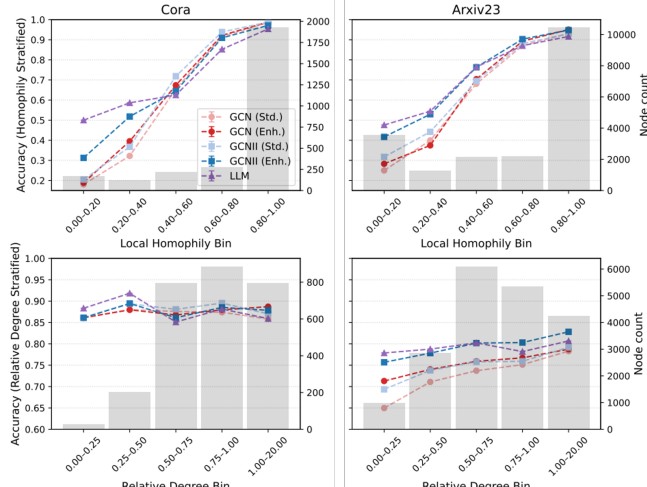

**Figure 1: Stratified performance for $h_v$ (top) and $\bar{d}_v$ (bottom).** Bars denote property distributions (right y-axis). Enhanced GNNs can benefit heterophilous & low degree nodes, but LLMs offer further improvements.

**Table 2:** (*Left*) NCS for $h_v$, $\hat{h_v}$, and $\bar{d_v}$ routing, where higher is better. Homophily yields high NCS, with estimated and true variants improving as $k$ increases. (*Right*) Mean rank across routing strategies, lower is better. **Bold** is best out of the label-free methods; we note that $h_v$ cannot be used during inference due to requiring access to labels.

| | | Cora | | | Pubmed | | | Arxiv23 | | | | | Avg. Ranking | Label Free For Inference |
|---|---|---|---|---|---|---|---|---|---|---|---|---|---|---|
| | *Strat.* | **10%** | **15%** | **20%** | **10%** | **15%** | **20%** | **10%** | **15%** | **20%** | | | | |
| | | | | | | | | | | | | Random | 4.50 | ✓ |
| **GCN Enh.** | $h_v$ | **0.24** | **0.11** | **0.05** | **0.29** | **0.30** | **0.26** | **0.15** | **0.14** | **0.15** | | C-density | 4.14 | ✓ |
| | $\hat{h_v}$ | -0.07 | -0.04 | -0.04 | 0.20 | 0.20 | 0.19 | 0.14 | **0.14** | **0.15** | | Degree | 4.33 | ✓ |
| | $\bar{d_v}$ | -0.07 | -0.07 | -0.06 | 0.05 | 0.04 | 0.04 | 0.03 | 0.05 | 0.03 | | Uncertainty | 3.28 | ✓ |
| | | | | | | | | | | | | $\bar{d_v}$ | 5.94 | ✓ |
| **GCNII Enh.** | $h_v$ | **0.09** | **0.03** | **0.00** | **0.10** | **0.11** | **0.09** | **0.04** | **0.05** | **0.06** | | $\hat{h_v}$ | **3.22** | ✓ |
| | $\hat{h_v}$ | -0.06 | -0.03 | -0.03 | 0.07 | 0.07 | 0.06 | **0.04** | 0.04 | 0.05 | | | | |
| | $\bar{d_v}$ | -0.06 | -0.07 | -0.05 | 0.03 | 0.02 | 0.02 | -0.03 | -0.01 | -0.01 | | $h_v$ | 1.03 | ✗ |

### 4.2.2 ROUTING WITH LOCAL HOMOPHILY

Unlike degree, which can be derived directly from the graph structure, homophily depends on class labels that are not accessible during training. This makes it challenging to directly exploit the performance gains LLMs provide on heterophilous nodes (Figure 1). Motivated by prior work on classifying edges as homophilous or heterophilous (Du et al., 2022; Wu et al., 2024), we extend this idea to estimate a node-level homophily score. Concretely, we train an MLP $Q$ to predict node labels, $\hat{y}_v = \arg\max Q(\mathbf{x}_v)$, and use these predictions to compute an estimated local homophily, $\hat{h}_v = \frac{1}{|\mathcal{N}(v)|} \sum_{u \in \mathcal{N}(v)} \mathbf{1}[\hat{y}_u = \hat{y}_v]$. This proxy enables us to leverage the benefits of homophily for routing without requiring ground-truth labels. We employ an MLP to avoid the structural biases that GNNs impose during homophily estimation (Zhu et al., 2021).

**Findings.** Table 2 shows that **homophily is a useful routing signal**. First, we find that true local homophily, $h_v$, attains the highest NCS in most settings, establishing an upper bound for structure-aware routing. More importantly, our *label-free* homophily estimate closely tracks $h_v$ and typically matches or surpasses other static heuristics. The average rankings in Table 2 corroborates this finding, where $h_v$ has the best mean rank for NCS score. Moreover, when $h_v$ is excluded from the ranking, our estimated homophily achieves the best average rank. Overall, homophily reliably identifies nodes that benefit from LLM assistance, and our label-free homophily proxy is a practical and effective prior for routing.

## 5 HOW TO ROUTE: ADAPTIVELY FUSING LLMS AND GNNS WITH GLANCE

To address the limitations of existing routing strategies, we propose **GLANCE**, a framework that adaptively fuses GNNs and LLMs. Rather than relying on handcrafted rules, GLANCE employs a lightweight router trained to decide, on a per-node basis, whether to invoke the LLM. This design ensures that the LLM is used only when it provides improvement to justify its cost. GLANCE is comprised of three components: (i) frozen GNN and LLM encoders, (ii) a trainable router using cheap features for routing, and (iii) a combiner that fuses structure and text embeddings into a final prediction.

### 5.1 COMPONENTS OF GLANCE

We now detail the components of GLANCE, as depicted in Figure 2. We first outline how nodes are chosen for routing. Then, we specify how the LLM generates new embeddings for routed nodes. Finally, we define the refinement process that merges the GNN and LLM information. As this is a non-differentiable pipeline, we include details on how the router is trained to encourage effective usage of the LLM.

### 5.1.1 STEP 1: GENERATING AND PROCESSING ROUTING FEATURES.

**Routing Features.** We begin by using a pre-trained GNN $F$ to produce embedding $\mathbf{z}_G(v)$ from the $k$-hop neighborhood of $v$. This embedding acts as the first signal to ensure the router can leverage the structural information of the neighborhood. We also use the GNN to derive uncertainty estimations on the node, following a dropout strategy (Qiao et al., 2025). This uncertainty acts as a proxy signal for difficulty, allowing the router to take advantage of the relationships found in Table 1. Building on the opportunity to leverage homophily, we utilize $Q$ to attain a local homophily estimate. However, to increase expressivity, we compute the probability distribution over the classes, $\mathbf{p}_{Q,v}$, and estimate a soft local homophily as:

$$\hat{h}_v = \mathbf{p}_{Q,v} \cdot \left( \frac{1}{|\mathcal{N}_1(v)|} \sum_{u \in \mathcal{N}_1(v)} \mathbf{p}_{Q,u} \right). \tag{1}$$

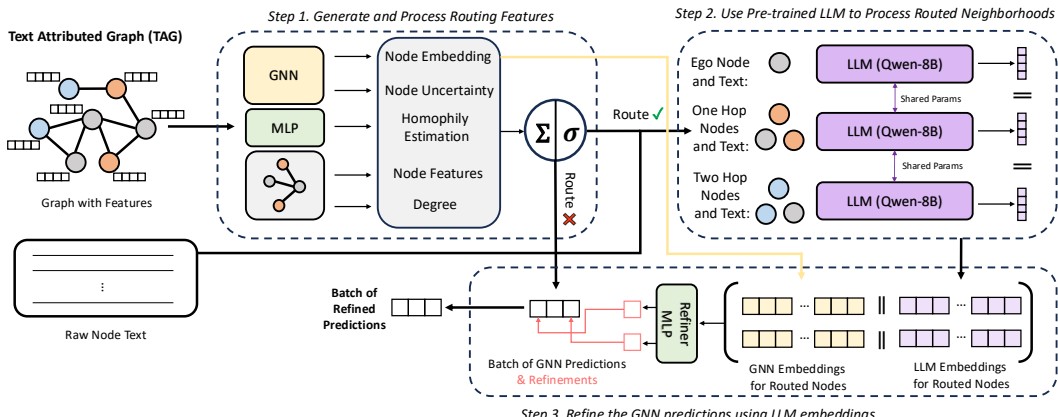

**Figure 2: GLANCE Overview.** *Step 1:* GLANCE generates routing features to derive a decision. *Step 2:* A routed node's text is fed into the LLM to generate embeddings. *Step 3:* A routed node's GNN & LLM embeddings are used to refine predictions. For nodes not routed, the GNN prediction is used. Only the router and refiner MLP are trained.

This soft estimate provides the router with a measure of neighborhood alignment. While Table 2 shows that $\hat{h}_v$ cannot definitively route on its own, we utilize this signal as a prior to bias the router toward heterophilous nodes, which we expect to be refined during training. While uncertainty was originally motivated to capture heterophily Qiao et al. (2025), we find this correlation to be weak (seen in Table 9), and use both metrics given they provide different, yet informative signals. Finally, we include the original features and degree, enabling the routing of nodes with noisy features or insufficient neighborhood context.

**Node Router.**   We define a router $\pi$ that takes as input the routing features above, denoted as a vector $\mathbf{f}_v$, and outputs a probability of routing $a_v \in [0,1]$, for a node $v$, as $a_v = \pi(\mathbf{f}_v) = \sigma(\mathbf{w}^\top \mathbf{f}_v)$, where larger values of $a_v$ indicate that the LLM should be queried for node $v$, while low values of $a_v$ indicates that the GNN prediction is sufficient for node $v$. Rather than applying an absolute threshold, GLANCE uses a top-$k$ strategy: for each mini-batch, the $k$ nodes with the highest $a_v$ scores are routed to the LLM. This ensures a fixed query budget per batch and avoids the need to globally calibrate the router's probabilities.

### 5.1.2   STEP 2: PRE-TRAINED LLM TO PROCESS ROUTED NEIGHBORHOODS

For the routed node set $R$, the LLM encoder $L$ is used to generate embeddings from serialized neighborhood prompts $\psi(\mathcal{N}_k(v)) \forall v \in R$, examples prompts given in Section B. Rather than producing a single embedding as in previous work (Wang et al., 2025), we generate embeddings at multiple structural levels: (1) the ego text $t_v$, (2) the ego text with a sampled set of 1-hop neighbors $\{t_v \cup t_u : u \in \mathcal{N}_1(v)\}$, (3) the ego text with a sampled set of 2-hop neighbors $\{t_v \cup t_u : u \in \mathcal{N}_2(v)\}$. Each level is serialized into separate prompts, encoded into embeddings $\mathbf{z}_{L,0}(v), \mathbf{z}_{L,1}(v), \mathbf{z}_{L,2}(v)$, and concatenated to form the final embedding:

$$\mathbf{z}_L(v) = [\mathbf{z}_{L,0}(v) \| \mathbf{z}_{L,1}(v) \| \mathbf{z}_{L,2}(v)].$$

This encoding preserves both ego and neighbor information, while ensuring the prompt length remains manageable. Additionally, this design aligns with the aggregation typically seen in advanced GNNs (Zhu et al., 2020; Yan et al., 2022). Using LLM embeddings also offers computational advantages, as it avoids the costly generation step required by prior approaches (Qiao et al., 2025; Wu et al., 2024).

### 5.1.3   STEP 3: REFINE THE GNN PREDICTIONS USING LLM EMBEDDINGS.

For nodes not routed, we retain their original GNN predictions by feeding the GNN-based node embeddings into the original MLP head used during GNN training. For nodes that are routed, we want to utilize both the GNN and LLM embeddings to capture structure and contextual information. Thus, we define a refiner MLP $C$ that integrates the GNN and LLM embedding as:

$$\mathbf{p}_{C,v} = \text{softmax}(C([\mathbf{z}_G(v) \| \mathbf{z}_L(v)])),$$

where $\mathbf{p}_{C,v}$ is the probability distribution over the classes for a node $v$. This modular design is agnostic to a specific GNN or LLM backbone, enabling flexibility depending on the dataset or computational budget.

## 5.2 TRAINING OBJECTIVES

Because routing decisions are discrete and require prompt construction, routing weights cannot be directly learned via backpropagation. Below, we detail how we translate the routing signal into weight updates.

**Counterfactual Comparison and Rewards.** When routing to the LLM, we measure the value of invoking it by computing a counterfactual prediction using the GNN and measuring the difference in losses produced by the two computations. Specifically, with $\mathbf{z}_G(v)$ and the GNN's pre-trained prediction head $H$:

$$\mathbf{p}_{H,v} = \text{softmax}(H(\mathbf{z}_G(v))),$$

$$\ell_v^{(GNN)} = -\sum_{k=1}^{|Y|}\mathbf{1}[y_v = k]\log\mathbf{p}_{H,v,k}, \quad \ell_v^{(LLM)} = -\sum_{k=1}^{|Y|}\mathbf{1}[y_v = k]\log\mathbf{p}_{C,v,k}$$

where $\ell_v^{(LLM)}$ is the loss when routing with the LLM, and $\ell_v^{(GNN)}$ is the counterfactual loss from the GNN. Note, the counterfactual is only computed when nodes are routed to the LLM during training, i.e. we do not compute an LLM-based counterfactual for nodes not routed. We then quantify the benefit of routing (or not routing) to the LLM through the rewards:

$$r_v = \begin{cases} \ell_v^{(GNN)} - \ell_v^{(LLM)} - \beta, & \text{if } a_v \text{ in top-}k \text{ (LLM queried)}, \\ -\ell_v^{(GNN)}, & \text{if } a_v \text{ not in top-}k \text{ (LLM not queried)}. \end{cases}$$

The first term captures the decrease in loss provided by querying the LLM and the cost of invoking the LLM (hyperparameter $\beta \geq 0$). Larger $\beta$ imposes stricter penalties, discouraging the use of the LLM where it doesn't provide benefit. A positive $r_v$ with LLM routing indicates that calling the LLM reduced prediction loss enough to offset its cost. When the LLM is not used, the reward is based on the GNN loss.

**Training Objective.** The router is optimized with a policy gradient-inspired loss, treating routing as a contextual bandit problem and encouraging alignment with counterfactual advantage:

$$\ell_v^{(route)} = -r_v\log\pi(\mathbf{f}_v) - \lambda_{\mathcal{H}}\mathcal{H}_{ent}[\pi(\mathbf{f}_v)].$$

While this objective is inspired by REINFORCE (Sutton et al., 1999), we employ deterministic top-$k$ selection during training to ensure stable use of the limited LLM query budget. The final objective jointly trains $C$ and $\pi$ through:

$$\ell_v^{(\text{pred})} = \mathbf{1}[a_v \in \text{top-}k]\ell_v^{(\text{LLM})} + \left(1 - \mathbf{1}[a_v \in \text{top-}k]\right)\ell_v^{(\text{GNN})}, \quad \mathcal{L} = \frac{1}{|\mathcal{B}|}\sum_{v \in \mathcal{B}}\ell_v^{(pred)} + \lambda_{\text{router}}\ell_v^{(route)}.$$

We use $\ell_v^{(pred)}$ to denote the prediction loss coming from either the LLM or GNN computation paths. Then, the first term of $\mathcal{L}$ optimizes predictive performance, while the second enforces cost-aware routing. For efficiency, both the GNN $F$ and the LLM $L$ are kept frozen during training, training only $C$ and $\pi$.

## 6 EMPIRICAL ANALYSIS

We now empirically study GLANCE, demonstrating how selectively using LLMs can improve performance. We first analyze GLANCE's performance, showing that it achieves significantly more balanced performance compared to the baselines. Then, we provide a series of supplemental analyses to understand what GLANCE learns during training. Finally, we extend GLANCE to larger TAGs to demonstrate its scalability.

**Table 3:** Per-bin accuracy for homophily levels. Training Strategies: **O** for Original features, **E** for Enhanced features, and L for LOGIN. **Bold** denotes best method. GLANCE achieves the strongest and most balanced performance across local homophily levels, as evidenced by its lowest average rank across datasets and homophily bins (rightmost column).

| | Strat. | Cora | | | | Pubmed | | | | Arxiv23 | | | | Avg rank |
|---|---|---|---|---|---|---|---|---|---|---|---|---|---|---|
| | | 0.00–0.25 | 0.25–0.50 | 0.50–0.75 | 0.75–1.00 | 0.00–0.25 | 0.25–0.50 | 0.50–0.75 | 0.75–1.00 | 0.00–0.25 | 0.25–0.50 | 0.50–0.75 | 0.75–1.00 | |
| GCN | O | 18.7±6.7 | 38.6±3.6 | 77.1±3.9 | 98.5±1.2 | 48.2±4.2 | 50.0±1.5 | 83.1±1.1 | 97.2±0.1 | 25.1±0.4 | 47.1±2.0 | 77.0±5.6 | 93.0±2.9 | 10.3 |
| | E | 17.9±6.9 | 49.6±6.7 | 78.5±3.3 | 98.7±0.3 | 55.9±3.7 | 51.8±3.8 | 84.9±0.8 | 97.9±0.1 | 27.8±3.0 | 46.4±2.0 | 79.1±4.4 | 95.1±2.1 | 8.0 |
| | L | 17.6±4.3 | 44.7±2.0 | 78.9±3.4 | 97.8±0.3 | 47.0±0.7 | 45.5±4.1 | 83.1±1.6 | **98.1±0.2** | 24.9±0.5 | 37.6±1.0 | 64.6±1.1 | 83.2±0.1 | 10.4 |
| SAGE | O | 21.1±2.9 | 45.2±3.3 | 76.2±4.8 | 97.9±0.5 | 58.4±2.7 | 66.1±1.4 | 85.3±0.6 | 96.6±0.2 | 32.6±0.3 | 53.3±0.4 | 80.5±0.7 | 93.9±0.4 | 8.6 |
| | E | 30.6±6.9 | 48.6±6.1 | 75.4±3.2 | 96.1±0.6 | 69.5±2.0 | **70.5±3.5** | 89.2±0.5 | 97.4±0.1 | 36.3±0.4 | 56.1±0.4 | 81.2±0.7 | 93.8±0.3 | 6.6 |
| | L | 26.8±2.6 | 43.6±5.0 | 75.4±3.8 | 97.8±0.4 | 67.1±1.3 | 69.6±1.8 | 88.5±0.9 | 97.6±0.4 | 37.0±2.7 | 51.6±3.1 | 76.0±1.1 | 89.8±1.0 | 8.2 |
| GCNII | O | 20.0±7.5 | 41.7±3.7 | 83.1±2.0 | **98.9±0.3** | 54.3±4.9 | 54.5±6.1 | 85.8±0.3 | 97.0±0.1 | 31.9±1.1 | 49.8±3.1 | 78.5±4.4 | 92.9±2.2 | 8.9 |
| | E | 32.0±7.9 | 53.5±3.5 | 77.6±1.5 | 97.1±0.9 | 69.7±2.1 | 68.1±4.6 | 88.9±0.9 | 97.4±0.4 | 41.6±1.2 | 58.0±2.7 | 83.2±1.9 | 94.8±0.7 | 4.7 |
| | L | 33.4±1.9 | 50.8±2.4 | 77.8±4.1 | 96.6±1.0 | 71.0±1.5 | 68.7±2.1 | 88.6±1.1 | 97.1±0.2 | **46.6±0.6** | 62.0±0.7 | 81.2±0.9 | 92.5±0.9 | 5.4 |
| FAGCN | E | 27.1±7.2 | 50.9±4.8 | **83.2±4.4** | **98.9±0.5** | 69.7±2.2 | 65.3±3.2 | 88.5±1.2 | 97.1±0.4 | 37.8±2.5 | 56.4±2.7 | 82.9±1.1 | 94.6±1.0 | 5.2 |
| GGCN | E | 27.7±8.4 | 50.2±9.5 | 71.4±1.8 | 95.8±0.9 | 68.8±1.6 | 66.0±2.8 | **89.4±0.5** | 97.7±0.1 | 42.3±0.5 | 62.1±0.4 | 85.0±0.2 | **95.3±0.3** | 5.3 |
| GBK-GNN | E | 28.6±4.8 | 51.0±2.1 | 74.4±2.8 | 96.7±0.8 | 67.5±1.9 | 67.9±4.3 | 87.9±1.7 | 97.9±0.0 | 38.7±0.9 | 58.8±1.2 | 83.8±1.2 | 94.3±0.1 | 5.8 |
| GLANCE | | **46.4±2.4** | **54.4±1.6** | 79.7±0.9 | 98.1±0.2 | **71.5±0.5** | 65.9±0.4 | **89.4±0.4** | 97.7±0.1 | 45.2±0.7 | **63.2±0.2** | **85.9±0.3** | **95.3±0.0** | **2.4** |

## 6.1 EXPERIMENTAL SETUP

**Datasets.** We first evaluate across Cora (Mccallum et al., 2000), Pubmed (Sen et al., 2008b), Arxiv23 (He et al., 2024), studying overall and stratified performance, as well as performing numerous sensitivity and ablation analyses. Then, to demonstrate GLANCE's scalability, we evaluate on two large-scale datasets, Arxiv-Year (Lim et al., 2021) and OGB-Products, with details provided in Section D. For each dataset, we follow the same training set up as outlined in the Section 4.1.

**Baselines & Training.** We compare GLANCE to a series of state-of-the-art baselines for scalable LLM-GNN fusion. We start by considering standard GNN backbones, including GCN (Kipf & Welling, 2017), GraphSAGE (Hamilton et al., 2017), and GCNII (Chen et al., 2020). We study their performance over three settings, original features, LLM-enhanced features, and LOGIN-filtered graphs. As GLANCE's core design is to bolster challenging heterophilous nodes, we also include high-performing heterophilous GNNs, including FAGCN, GGCN, and GBK-GNN. To imbue them with LLM knowledge, we utilize LLM-enhanced features. We do not apply LOGIN on these models as they are designed to utilize heterophily, whereas LOGIN's design intends to remove it. Across all baselines, we use identical data splits, training protocols, and text encoders. Training details and hyperparameters are provided in Section C. Unless otherwise stated, we route the top 12 nodes per batch of 32 nodes for GLANCE for reported metrics. For OGB-Products, we evaluate against the top 3 model combinations found in our initial study.

## 6.2 CAN GLANCE ACHIEVE THE BENEFITS OF LLMs AND GNNs IN ONE MODEL?

As shown in Figure 1, selectively using the LLM on challenging nodes can enhance GNN performance. To evaluate GLANCE's capabilities, we report both overall accuracy (Table 4) and stratified results (Table 3). GLANCE attains the best performance on all three datasets, achieving 89.5% on Cora, 92.6% on Pubmed, and 82.1% on Arxiv23, leading to an average gain of +0.5% over the next best model. Notably, GLANCE requires significantly fewer LLM queries to attain similar performance gains found in previous work (Wang et al., 2025; Chen et al., 2024a). GLANCE also consistently outperforms LLM-enhanced and LOGIN baselines, often by several percent, as well as GNNs designed for heterophily. From a stratified perspective, the largest improvements emerge on heterophilous nodes, where GLANCE delivers +13% on Cora and +0.5% on Pubmed. Importantly, when averaging across bins, **GLANCE achieves the strongest overall balance with an average rank of 2.4, compared to 4.7 for the next best**. Together, these findings demonstrate that GLANCE's routing reliably leverages the LLM where it is most beneficial while preserving the strengths of the GNN.

**Table 4:** Acc for 3 runs. **Bold** is best

| | | Cora | Pubmed | Arxiv23 |
|---|---|---|---|---|
| GCN | O | 87.1±1.9 | 88.1±0.6 | 74.2±2.5 |
| | E | 87.9±1.3 | 89.9±0.6 | 76.2±1.2 |
| | L | 86.8±0.4 | 88.6±0.2 | 65.7±0.0 |
| SAGE | O | 86.9±1.0 | 89.7±0.4 | 77.2±0.2 |
| | E | 86.4±1.1 | 92.2±0.4 | 78.1±0.1 |
| | L | 86.4±0.3 | 91.8±0.1 | 74.7±1.5 |
| GCNII | O | 88.4±0.8 | 89.1±0.6 | 75.9±2.1 |
| | E | 87.7±1.3 | 92.1±0.2 | 80.2±0.7 |
| | L | 86.8±0.7 | 91.9±0.1 | 79.7±0.5 |
| FAGCN | E | 89.4±0.5 | 91.8±0.2 | 79.2±1.3 |
| GGCN | E | 85.4±0.9 | 92.2±0.2 | 81.2±0.5 |
| GBK-GNN | E | 86.6±1.0 | 92.1±0.3 | 79.5±0.4 |
| GLANCE | | **89.5±0.4** | **92.6±0.1** | **82.1±0.1** |

**Properties of Routed Nodes.** In Figure 3, we analyze the local homophily of routed nodes, splitting them into benefited (GNN wrong, LLM right) and non-benefited sets. Across datasets, we observe substantial mass at low homophily, showing that GLANCE preferentially routes heterophilous nodes. Crucially, the benefit distribution is skewed toward low homophily, indicating that routed heterophilous nodes deliver performance gains. This supports our hypothesis that **heterophily captures GNN failure modes where LLMs are most valuable**, and shows that these nodes drive GLANCE's improvement. We also highlight that the median beneficial homophily value is not identical across datasets, e.g., on Arxiv23 the largest gains occur around $h \approx 0.5$. Thus, **a single homophily threshold that only routes heterophilous nodes would be ineffective for maximizing performance**, and additional context is needed.

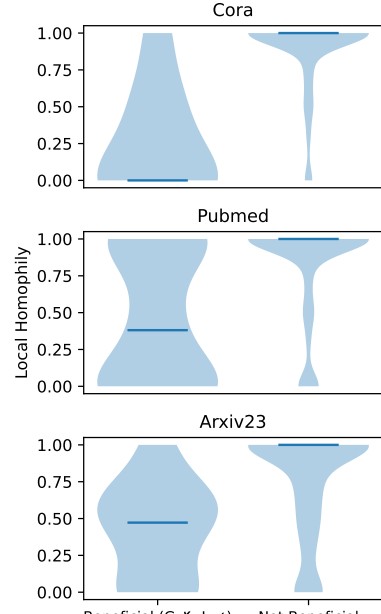

**Figure 3: Local homophily for routed nodes**, split by benefit. Blue lines denote median.

**Sensitivity to $K$.** In Figure 6 (Section F.1), we examine performance under varying routing budgets. For a batch size of 32, we evaluate $K \in \{8, 12, 16\}$. We find that increasing $K$ tends to lead to steady gains on heterophilous nodes, while leaving performance on the homophilous nodes mostly unchanged. On Pubmed and Arxiv23, accuracy on $h_v < 0.25$ improves by $+3.4\%$ for $K = 8 \to 12$, and by another $+3.0\%$ for $K = 12 \to 16$. On Cora, performance initially dips for $K = 8 \to 12$, but then significantly improves by $+12.3\%$ when increasing to $K = 16$. For $h_v > 0.75$, GLANCE's accuracy drops by a marginal $-0.06\%$ on average across datasets, underscoring GLANCE's stability in this regime. Importantly, we also observe that routing too many nodes can harm accuracy (as seen in Table 10 for full-batch routing), as easy nodes well modeled by the GNN can become misclassified when the LLM is unnecessarily used. Together, these findings demonstrate that **GLANCE routes efficiently, mitigating both the performance degradation and computational overhead of indiscriminate LLM usage.**

## 6.3 ROUTER: LEARNED PROPERTIES AND ROBUSTNESS

**Routing Feature Ablation.** To assess the value of the routing features, we disable each feature one at a time and train new GLANCE models, measuring $\text{Acc}_{abl} - \text{Acc}_{full}$ (Figures in Section F.3). Averaged over all features, accuracy drops on each dataset, with changes of $-0.38\%$ on Cora, $-1.07\%$ on Pubmed, and $-0.65\%$ on Arxiv23. More importantly, when ablating the homophily feature, we find significant drops when $h_v < 0.5$, with changes of $-6.5\%$ for Cora, $-6.3\%$ for Pubmed, and $-2.0\%$ for Arxiv23. We also observe subtle losses when $h_v > 0.5$ as the homophily signal is removed, where Cora drops by $-0.34\%$, Pubmed drops by $-0.15\%$, and Arxiv23 drops by $-0.03\%$. These drops indicate that homophily helps the router avoid misrouting the nodes easy for GNNs. Together, these ablations show that **GLANCE achieves peak performance when all features are available**, with the largest gains on heterophilous nodes.

## 6.4 LARGE-SCALE LEARNING WITH GLANCE

**Datasets.** GLANCE's selective utilization of the LLM enables the processing of large TAGs by only applying the LLM to nodes that benefit. Now that we have demonstrated the effectiveness of GLANCE, we study the scalability by applying GLANCE to two large benchmark datasets, Arxiv-Year (Lim et al., 2021) and OGB-Products (Hu et al., 2020). Details for these datasets can be found in Section D. Notably, Arxiv-Year and OGB-Products are multiple orders of magnitude larger than the datasets typically studied in GNN-LLM papers (Qiao et al., 2025; Wu et al., 2024; Wang et al., 2025). Furthermore, to motivate GLANCE's scalability, we show in Figure 9 that the routing and refinement components of GLANCE create very little overhead, enabling it to handle larger datasets when querying less nodes with the LLM.

**Results.** In Table 5, we provide per-bin and overall accuracy for the top 3 average rank models from Table 3 and compare them to GLANCE. Given these datasets are significantly larger than the previous datasets, we utilize a query rate of $\sim 1.6\%$ ($K = 1$ with batch size 64). We find that GLANCE outperforms

**Table 5:** Per-bin accuracy for homophily levels and overall performance on larger datasets. We compare GLANCE to the next 3 best ranking models from Table 3, all using the enhanced feature strategy (E). OOM for GGCN discussed in Section C.1. **Bold** denotes best. We find the benefits carry to large-scale datasets with higher overall performance.

| | | Arxiv-Year | | | | | OGB-Products | | | | |
|---|---|---|---|---|---|---|---|---|---|---|---|
| | | 0.00–0.25 | 0.25–0.50 | 0.50–0.75 | 0.75–1.00 | Overall | 0.00–0.25 | 0.25–0.50 | 0.50–0.75 | 0.75–1.00 | Overall |
| GCNII | E | 42.7±0.0 | 51.1±0.0 | **54.5±0.0** | **72.4±0.1** | 49.6±0.1 | 29.0±0.1 | 43.9±0.3 | 69.3±0.0 | 91.9±0.1 | 81.8±0.1 |
| FAGCN | E | 36.0±0.9 | 45.0±0.5 | 52.7±0.2 | 72.0±0.5 | 44.3±0.5 | **29.8±1.8** | 42.3±1.3 | 64.7±1.1 | 88.3±1.4 | 78.6±0.9 |
| GGCN | E | 36.8±0.5 | 46.3±0.6 | 51.6±1.4 | 69.3±1.2 | 44.6±0.1 | OOM | OOM | OOM | OOM | OOM |
| GLANCE | | **43.1±0.1** | **51.3±0.1** | 54.2±0.0 | 72.1±0.1 | **49.8±0.1** | 29.7±0.3 | **44.4±0.0** | **69.9±0.1** | **92.3±0.0** | **82.3±0.1** |

the best baselines while achieving strong performance across the spectrum of homophily levels. These results demonstrate that selective LLM queries, even at very low query rates, can enable gains and supplement GNNs where they struggle. Given the results seen above with regards to sensitivity to $K$, we also expect this performance can become even better with a larger compute budget. Moreover, for applications that necessitate better performance on heterophilous pockets of nodes, GLANCE is able to offer key benefits, regardless of dataset size, that are otherwise inaccessible with standard GNN architectures.

## 7 CONCLUSION

In this work we focused on how LLMs can be selectively utilized to bolster GNNs, an understudied area in the LLM-GNN literature. Through a detailed analysis of structural properties, we showed that homophily provides a reliable signal for routing, capturing where GNNs tend to fail and LLMs excel. To operationalize this finding, we proposed GLANCE, a cost-aware fusion strategy that learns when to route, calling the LLM on difficult nodes. Empirical results across multiple benchmarks demonstrate that GLANCE achieves well-balanced performance while selectively invoking the LLM only when useful. Our findings argue for structure-aware routing as a foundation for future work on efficient GNN-LLM integration.

## 8 ACKNOWLEDGMENTS

The material in this work is supported by the National Science Foundation under IIS 2504090, IIS 2212143, and CAREER Grant No.IIS 1845491. Any opinions, findings, and conclusions or recommendations expressed in this material are those of the author(s) and do not necessarily reflect the views of the National Science Foundation or other funding parties.

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

## A    DETAILED RELATED WORK

**Static GNN-LLM Fusion.**    Recent work has focused on two main paradigms for graph learning with LLMs: *LLMs-as-Enhancers* and *LLMs-as-Predictors* (Chen et al., 2024a; Li et al., 2024). For LLM-as-Enhancers, LLMs generate enriched node features, ranging from simple text embeddings (Wu et al., 2024) to additional external knowledge (He et al., 2024; Chen et al., 2024a), effectively performing automated feature engineering to bolster GNN performance. For LLM-as-Predictors, the LLM replaces the GNN as the final classifier, processing serialized graph structure and text as prompts (Wu et al., 2024; Wang et al., 2025). Some variants retain a GNN encoder (Lin et al., 2025), but the LLM still produces the prediction. While both paradigms benefit from the reasoning of LLMs, they also suffer from distinct limitations. Enhancer methods are limited by their static, frozen features while also inheriting the inductive biases of the GNN, retaining the challenges seen with certain structural properties (e.g., heterophily). In contrast, LLM-as-Predictor methods can overcome the GNN inductive biases, but may also struggle to encode topology efficiently given prompt length limits and serialization mismatches (Firooz et al., 2025; Wang et al., 2025). Another common approach to fusing models together is through Mixture of Experts (MoE) which ensemble a set of models, each with expert knowledge. While prominent in LLM design (Cai et al., 2025) and graph MoE Wang et al. (2023), this strategy has yet to be applied to a mixture of LLMs and GNNs. Relatedly, LLMs have been shown capable to route to different GNNs (Jiang & Luo, 2025), yet, this still limits the pipeline to the inductive biases of the GNNs. Our approach adopts a hybrid paradigm that selectively invokes the LLM, using expert knowledge from both models, only when it is expected to help performance.

**Adaptive GNN-LLM Fusion.**    A challenge for LLM-GNN models is high inference cost, especially when LLMs are used during training. While the static methods above are scalable by treating the LLM as a pre-processer, many methods consider using the LLM within their forward pass, calling the LLM across training and inference (Wu et al., 2024). While attempts to reduce LLM cost via neighborhood sampling or small language model distillation show promise (Fang et al., 2023; Chen et al., 2024a), the bottleneck remains where the LLM must be applied to all nodes. Only a few recent works explore adaptive GNN-LLM fusion, where the LLM is invoked selectively rather than universally. E-LLaGNN is one of the first in this space, selecting nodes for LLM augmentation using fixed heuristics based on graph properties such as degree, centrality, or text length (Jaiswal et al., 2024). While effective in some settings, these handcrafted rules require manual tuning and can vary in effectiveness across datasets. LOGIN (Qiao et al., 2025) introduces a more automated approach, using GNN uncertainty to identify challenging nodes to rewire and simplify message passing. However, this rewiring can disrupt structurally informative but challenging patterns, such as heterophilous links, which can be beneficial in real-world graphs (Luan et al., 2021). LLM-GNN is a label-free approach that leverages a similar principle as LOGIN, but uses clustering density as a proxy of difficulty as opposed to uncertainty (Chen et al., 2024b). Our approach differs from these methods in three key ways. First, while different metrics have been proposed to route, our systematic study identifies specific structural characteristics that benefit from LLM processing. Additionally, GLANCE does not require prespecified thresholds on these properties, instead learning how to utilize them to route. Second, instead of editing the underlying data to favor GNNs, we preserve its characteristics and invoke the LLM where the GNN struggles. Third, while LOGIN and LLM-GNN address non-differentiability by editing the underlying data to provide signal from the LLM, we retain the true graph signal and directly train the routing policy.

## B    LARGE LANGUAGE MODEL PROMPTING

In this section, we detail how we generate enhanced features for the base GNN models. Additionally, we provide prompts used to generate LLM embeddings within GLANCE. For both settings, we focus on the Qwen3 family of models, specifically using the Qwen3-Embed-8B model to generate embeddings.

### B.1    LLM-AS-ENHANCER PROMPTS

We generate node-level text embeddings using Qwen3-Embed-8B by directly processing the node text (e.g., title and abstract for Cora). We use a max prompt length of 1024. Following the model's official configuration, we use last-token pooling and then apply $\ell_2$ normalization. Concretely, for a batch of tokenized sequences with hidden states $H \in \mathbb{R}^{B \times L \times d}$ and (unpadded) last-token indices $\ell_i$, the sentence

embedding for node $i$ is

$$\tilde{z}_i = H_{i,\ell_i} \in \mathbb{R}^{4096}, \qquad z_i = \frac{\tilde{z}_i}{\|\tilde{z}_i\|_2}.$$

We keep Qwen's include_prompt behavior enabled, which prepends a short instruction prompt internally before the text. The resulting $z_i$ is used as an enhanced feature in place of the shallow features outlined above.

### B.2 LOGIN PROMPTS

Following the format introduced in LOGIN, we construct prompts for each node by including the text of the ego node, a template that describes the graph and the instructions for the LLM, and the ground-truth labels and GNN predicted labels for nodes in the one-hop and two-hop neighborhood of the ego node.

**LOGIN Example for Cora:**

```
NODE 0: <ego-node text>

Given a citation graph
    NODE-IDX: <node_id>,
    NODE-LIST: [List of all node_id in two-hop neighborhood],
    ONE-HOP-NEIGHBORS: [List of node_id of one-hop neighbors],
    TWO-HOP-NEIGHBORS: [List of node_id of two-hop neighbors]
    NODE-LABELS: [List of ground truth labels]
    GNN-PREDICTED-NODE-LABEL: <GNN-predicted ego node label>
where node 0 is the target paper
and you see the true labels under 'node_label'.

Question: Which category does this paper belong to?
Choose exactly one from [list of class labels].

Return JSON
{{classification result: <choice>,
  explanation: <your reasoning>}}
</END>
```

### B.3 LLM-AS-PREDICTOR AND GLANCE PROMPTS

When using the LLM as predictor, we construct prompts for different hops of the neighborhood (ego-only, ego+1-hop, ego+2-hop) and encode them with Qwen3-Embed-8B. Doing this enables the LLM to retain per-hop information, a key design found in heterophilous graph learning, while also helping keep prompt lengths manageable. We utilize the same parameterizations as the LLM-as-Enhancer setting for each of these embeddings, but use a prompt length of 512 for the ego node, and 4096 for the one- and two-hop embeddings. Each prompt is formatted in a instruction–query layout compatible with Qwen3 embedding prompts and utilizes last-token pooling of the sequence.

The overall strategy in prompt design was to provide the classes and serialized neighborhood structure for a node. We do not over-engineer the prompt design, however, this can be one area for further opportunity in graph learning with LLMs. Additionally, providing additional context on a per-dataset basis (e.g. highlighting the domain of the dataset), could provide additional value.

**Ego-Only Example:**

```
Instruct: Predict the node's category from the provided context.
Possible categories: [list of class labels].
Query:
EGO:
<ego-node text>
Category?
</END>
```

**Ego+1 Hop Example:**

```
Instruct: Predict the node's category from the provided context.
Possible categories: [list of class labels].
Query:
EGO:
<ego-node text>
HOP1:
- <1 hop neighbor 1 text>
- <1 hop neighbor 2 text>
...
Category?
</END>
```

**Ego+2 Hop Example:**

```
Instruct: Predict the node's category from the provided context.
Possible categories: [list of class labels].
Query:
EGO:
<ego-node text>
HOP2:
- <2 hop neighbor 1 text>
- <2 hop neighbor 2 text>
...
Category?
</END>
```

As the degree of the datasets can be large, we use a neighbor sampling strategy during prompt construction to further maintain manageable prompt sizes. Specifically, we limit the size of each neighborhood to be up to 5 neighbors per node. Thus, the ego-only text contains one node, the ego + 1-hop can contain up to 6 nodes (1 + 5 neighbors), and the ego + 2-hop can contain up to 26 nodes (1 + 5*5). We use a prompt length of 1024 for the ego-node, and 4096 for the ego + 1-hop and ego + 2-hop. On a per-node basis, text is capped to a fixed character budget (2000 characters). We process the embeddings using $\ell_2$ normalization before concatenation.

When originally fine-tuning the LLM (such as the results in Figure 1, we apply a 2-layer MLP to map the concatenated embeddings to the label space for each model. Then, when incorporating the pre-trained LLM into GLANCE, we remove the MLP head and use the concatenated embeddings directly with the refiner MLP.

## C  MODEL TRAINING AND HYPERPARAMETERS

We describe here the training protocols for the base GNNs, the LLM-as-Predictor baseline, and our proposed GLANCE model. Across all models, we tune the learning rate from $\{0.01, 0.001, 0.0001\}$ and the weight decay from $\{0.001, 0.0001, 0.00001\}$. We apply gradient clipping at $\|g\|_2 \leq 1.0$ for all gradients $g$. Training is performed on a single NVIDIA A40 GPU using the AdamW optimizer, with dataset splits of 50/25/25 for train/validation/test. For OGB-Products, we cap the train set to 50,000 nodes for GLANCE, but retain the same test set with the GNN models.

### C.1  GNN (AND MLP) TRAINING

For the GNN backbones, we use implementations integrated into PyTorch Geometric when available. Otherwise, for GBK-GNN [1] and GGCN[2] we use their official implementations. For all GNNs, we use a hidden dimension of 64 and a batch size of 128. We search over network depths of $\{2, 3\}$ layers. Training is run for up to 1000 epochs, with early stopping applied if the validation accuracy does not improve for 30 consecutive epochs. Each model includes a one-layer MLP projection head, which both

---

[1] https://github.com/Xzh0u/GBK-GNN
[2] https://github.com/Yujun-Yan/Heterophily_and_oversmoothing

produces the final predictions and allows the learned GNN embeddings to be reused within GLANCE. For heterophilous architectures such as FAGCN, GGCN, and GBK-GNN, we adopt the hyperparameter settings recommended in their official repositories. For the MLP local homophily estimator, we follow a similar strategy as the GNNs, using a depth of 2 and hidden dimension of 64. Both the GNN and MLP additionally use ReLU activations and a dropout of 0.2. For OGB-Products, we limit to 20 epochs with a patience of 5, and limit the neighbor sampling rate to 30 nodes per neighbor. Additionally, we highlight that the official implementation of GGCN runs out of memory for OGB-Products due to requiring access to the full graph upon initialization. Even with sparse tensors, this is found to be impossible for the OGB-Products dataset.

## C.2    LOGIN TRAINING

For LOGIN, we use GNN backbones with setups described in the previous section. We then estimate the uncertainty of nodes by doing five forward passes with random dropout at a $0.3$ dropout rate. We take the top $k$ nodes with $k$ defined by:

$$k = \min(1000, n(1 - h_v))$$

to query the LLM, where $n$ is the number of nodes and $1 - h_v$ is the *heterophily* ratio. While LOGIN originally utilizes the heterophily ratio to query nodes, we find this to be too large for Arxiv23 and causes large training times. Thus, the 1000 node cap allows us to utilize LOGIN for Arxiv23. We use a pretrained Vicuna-7B to generate the predictions and explanations. Depending on whether the LLM predict labels match the ground-truth labels, we either append the explanation to the original node text or prune edges in the node's one-hop neighborhood as implemented in LOGIN. We use a similarity threshold of $0.15$ for pruning the edges of nodes whose LLM prediction does not match the ground-truth label. Finally, a pretrained E5-Large model is used to encode the new set of node text that includes the original text and LLM explanations.

## C.3    LLM TRAINING

For the LLM baseline, we fine-tune Qwen3-Embed-8B to serve as an LLM-as-Predictor. To reduce both memory and compute, we employ parameter-efficient adaptation via Low-Rank Adaptation (LoRA), inserting adapters into the attention projection layers with a rank of 16 and scaling factor $\alpha = 32$. We also enable FlashAttention-2 and use bfloat16 precision. During training, only LoRA parameters are updated while the base model remains frozen.

We use a batch size of 1 with gradient accumulation over 8 steps, yielding an effective batch size of 8. Training proceeds for a maximum of 10 epochs with early stopping applied if validation accuracy fails to improve for 2 epochs. Since training on all available nodes is infeasible, we cap the training set at 3,000 randomly sampled nodes. As with the GNNs, we include a one-layer MLP head to project embeddings into the label space.

## C.4    GLANCE TRAINING

When training GLANCE, we first independently train a GNN, LLM, and shallow MLP via the training processes explained in Section C.1 and Section C.3. Specifically, each backbone model is trained to optimize the node classification task and is then frozen to ensure its modeling capabilities are maintained within GLANCE. Thus, when training GLANCE, only the *router* and the *refiner* models that fuse the two pathways are updated. The router is implemented as a logistic regression model to predict the likelihood of routing, while the refiner is a 2-layer MLP following the same configuration as the MLP local homophily estimator. Routing decisions are made by selecting the top-$K$ nodes per batch according to the router's predicted scores. To gradually reduce reliance on the LLM during training, we decay the routing budget across epochs $t = 1, 2, ...$ using an exponential schedule:

$$K_t = \text{round}\big(K_{\text{end}} + \big(K_{\text{start}} - K_{\text{end}}\big) r^{t-1}\big),$$

where $K_{\text{start}}$ equals the batch size, $K_{\text{end}}$ is the ending routing budget for the last epoch, and $r$ is the decay factor. We set $K_{end} = K_{\text{start}}/4$ and $r = 0.5$ during our experiments, requiring significantly less LLM calls across training.

We train with mixed-precision using bfloat16, and since no gradients are propagated through the LLM, the memory footprint is greatly reduced, allowing us to use a batch size of 32. For the different datasets, we

set $K_{\text{start}}$ equal to the batch size. As in the LLM baseline, we cap the training set at 3,000 nodes. Early stopping is applied with a patience of 2 epochs. We additionally tune the LLM query cost $\beta \in \{0.1, 0.2, 0.3\}$, and set the router weight $\lambda_{\text{router}} = 1.0$ and entropy regularization weight $\lambda_{\text{ent}} = 0.01$ by default.

## D  DATASET DETAILS

To characterize the limitations of current heuristics to route, as well as evaluate GLANCE, we utilize three widely studied TAG benchmarks: Cora (Mccallum et al., 2000), Pubmed (Sen et al., 2008a), and Arxiv23 (He et al., 2024). Then, for our study on scalability, we utilize OGB-Products (Hu et al., 2020), representing a multiple order of magnitude increase in dataset size compared to typically studied TAGs. These datasets also differ significantly in homophily levels, allowing us to probe routing decisions under diverse conditions.

- **Cora.** Cora is a citation network consisting of machine learning papers. Papers share an edge if they share a citation. The data is from `https://github.com/myflashbarry/LLM-benchmarking` (Wang et al., 2025). The canonical feature vector for each paper is a bag-of-words feature vector. The nodes are categorized into 7 different AI subfields. Cora possesses the highest global homophily level out of the datasets ($h \approx 0.81$).

- **Pubmed.** Pubmed is a larger citation graph on biomedical articles, where papers share an edge if they share a citation. The data is from `https://github.com/myflashbarry/LLM-benchmarking`. Each article is represented by a TF–IDF word vector, with labels corresponding to 3 different disease types. Similar to Cora, Pubmed exhibits high global homophily ($h \approx 0.80$), but also possesses a larger pocket of heterophilous nodes (see Figure 1).

- **Arxiv23.** Arxiv23 is a recently curated citation subgraph of arXiv covering computer science papers published after 2023, surpassing the knowledge cutoff of flagship models like ChatGPT-3.5. The data is from `https://github.com/XiaoxinHe/tape_arxiv_2023`. We find that the original proposed Arxiv23 dataset has a large number of isolated nodes, which can confound the evaluation metrics given over half of the nodes do not possess a graph structure. Thus, we subset Arxiv23 to only include the largest connected component. Each article is represented by a Word2vec embedding, with each class corresponding to a computer science subfield. Unlike Cora and Pubmed, Arxiv23 displays a wider array of homophilous and heterophilous edges, with ($h \approx 0.67$), making it a challenging benchmark for neighborhood-averaging GNNs.

- **Arxiv-Year** Arxiv-Year is a heterophilous variant of OGB-Arxiv, using the publication year as the label. The data is from `https://ogb.stanford.edu`, but with the labels created via quantile ranges (see `https://github.com/CUAI/Non-Homophily-Large-Scale/blob/master/data_utils.py`, producing 5 labels. The dataset is roughly an order of magnitude larger compared to the three previous datasets. For this dataset, we only use enhanced features from the raw text. Arxiv-year is heterophilous ($h \approx 0.22$).

- **OGB-Products** OGB-Products is a large-scale e-commerce TAG provided from the OGB `https://ogb.stanford.edu`. The task is to predict the category of e-commerce items, where items share a link if co-purchased. The dataset is roughly two orders of magnitude larger than the TAGs seen in previous literature, with 47 classes. We only use enhanced features for this dataset. OGB-Products is homophilous ($h \approx 0.81$)

The explicit number of features, classes, and other key metrics are summarized in the table below. Notably, the datasets we study contain a wide variety of sizes, densities, and homophily levels.

**Table 6:** Summary statistics of the datasets used in our experiments.

| Dataset | #Nodes | #Edges | Original Feature Dim | #Classes | Global $h$ |
|---|---|---|---|---|---|
| Cora | 2,708 | 5,278 | 1433 | 7 | 0.81 |
| Pubmed | 19,717 | 44,324 | 500 | 3 | 0.80 |
| Arxiv23 | 19,550 | 55,350 | 300 | 40 | 0.67 |
| Arxiv-Year | 169,343 | 1,166,243 | - | 5 | 0.22 |
| OGB-Products | 2,449,029 | 61,859,140 | - | 47 | 0.81 |

**Table 7:** NCS for difficulty-, $d_v$-, and uncertainty-based routing is compared against a random baseline which randomly routes nodes to the LLM. Higher NCS is better. Numbers in parenthesis denotes number of nodes routed.

| | | Cora | | | Pubmed | | | Arxiv23 | | |
|---|---|---|---|---|---|---|---|---|---|---|
| | | **10%** | **15%** | **20%** | **10%** | **15%** | **20%** | **10%** | **15%** | **20%** |
| | *Routing Strat.* | (68) | (102) | (136) | (493) | (740) | (986) | (489) | (734) | (978) |
| **GCN** | Random | -0.029 | -0.01 | -0.029 | 0.049 | 0.062 | 0.062 | 0.145 | 0.083 | 0.092 |
| | C-density | -0.044 | -0.069 | -0.074 | 0.065 | 0.07 | 0.065 | 0.096 | 0.091 | 0.092 |
| | Degree ($d_v$) | -0.029 | -0.02 | -0.037 | 0.059 | 0.059 | 0.064 | 0.084 | 0.094 | 0.105 |
| | Uncertainty | -0.147 | -0.088 | -0.044 | 0.183 | 0.15 | 0.141 | 0.139 | 0.144 | 0.14 |
| **GCNII** | Random | -0.044 | -0.069 | -0.029 | 0.041 | 0.028 | 0.053 | 0.078 | 0.038 | 0.058 |
| | C-density | -0.015 | -0.049 | -0.044 | 0.041 | 0.045 | 0.045 | 0.08 | 0.079 | 0.077 |
| | Degree ($d_v$) | -0.029 | -0.02 | -0.029 | 0.059 | 0.055 | 0.053 | 0.063 | 0.068 | 0.078 |
| | Uncertainty | -0.147 | -0.088 | -0.051 | 0.132 | 0.107 | 0.117 | 0.117 | 0.134 | 0.127 |

**Table 8:** NCS for estimated and true homophily routing strategies with different routing top-$k$ percentages. Higher NCS is better. Numbers in parenthesis denotes number of nodes routed. Homophily is shown to produce high NCS values, with both the estimated and true variants producing larger NCS values as $k$ increases.

| | | Cora | | | Pubmed | | | Arxiv23 | | |
|---|---|---|---|---|---|---|---|---|---|---|
| | | **10%** | **15%** | **20%** | **10%** | **15%** | **20%** | **10%** | **15%** | **20%** |
| | *Routing Strat.* | (68) | (102) | (136) | (493) | (740) | (986) | (489) | (734) | (978) |
| **GCN** | $h_v$ | 0.265 | 0.118 | 0.081 | 0.314 | 0.299 | 0.266 | 0.217 | 0.217 | 0.212 |
| | $\hat{h_v}$ | -0.059 | -0.078 | -0.074 | 0.215 | 0.199 | 0.193 | 0.196 | 0.192 | 0.187 |
| | $\bar{d_v}$ | -0.059 | -0.069 | -0.051 | 0.055 | 0.041 | 0.038 | 0.02 | 0.059 | 0.057 |
| **GCNII** | $h_v$ | 0.235 | 0.098 | 0.066 | 0.213 | 0.2 | 0.168 | 0.176 | 0.159 | 0.15 |
| | $\hat{h_v}$ | -0.074 | -0.069 | -0.074 | 0.154 | 0.131 | 0.123 | 0.137 | 0.114 | 0.122 |
| | $\bar{d_v}$ | 0.0 | -0.039 | -0.037 | 0.045 | 0.026 | 0.022 | 0.033 | 0.064 | 0.056 |

# E  ROUTING FOR GCN AND GCNII

In addition to our enhanced results provided in the main section of the paper, we include supplemental results with routing on the base GCN and GCNII architectures with their original features.

## E.1  ROUTING WITH PRIOR METHODS

Similar to the results seen on the enhanced model variants, we see that in Table 7, across Cora, Pubmed, and Arxiv23, uncertainty-based routing consistently delivers the strongest NCS among the heuristic baselines for both backbones, with degree and C-density lagging behind (often near or below the random baseline on Cora). This result confirms our original finding that, when routing into an LLM, predictive uncertainty is the most reliable heuristic from the prior methods, while the other signals alone can even harm performance.

## E.2  HOMOPHILY-BASED ROUTING (ESTIMATED VS. TRUE)

We begin by evaluating the effectiveness of the true local homophily, $h_v$. We can see in Table 8, similar to the main text, $h_v$ achieves the highest NCS across models and datasets. However, when we move to estimated homophily $\hat{h_v}$, we can see that it follows a similar relative trend. Relative degree $\bar{d_v}$ maintains its relatively weaker performance. These findings further confirm that homophily is a powerful routing signal, with $\hat{h_v}$ conferring robust gains.

**Table 9:** Correlation between model uncertainty and (true) homophily.

| Dataset | Corr(uncertainty, $\hat{h}_v$) | Corr(uncertainty, $h_v$) |
|---|---|---|
| Cora | -0.468 | -0.334 |
| Pubmed | -0.383 | -0.331 |
| Arxiv23 | -0.407 | -0.381 |

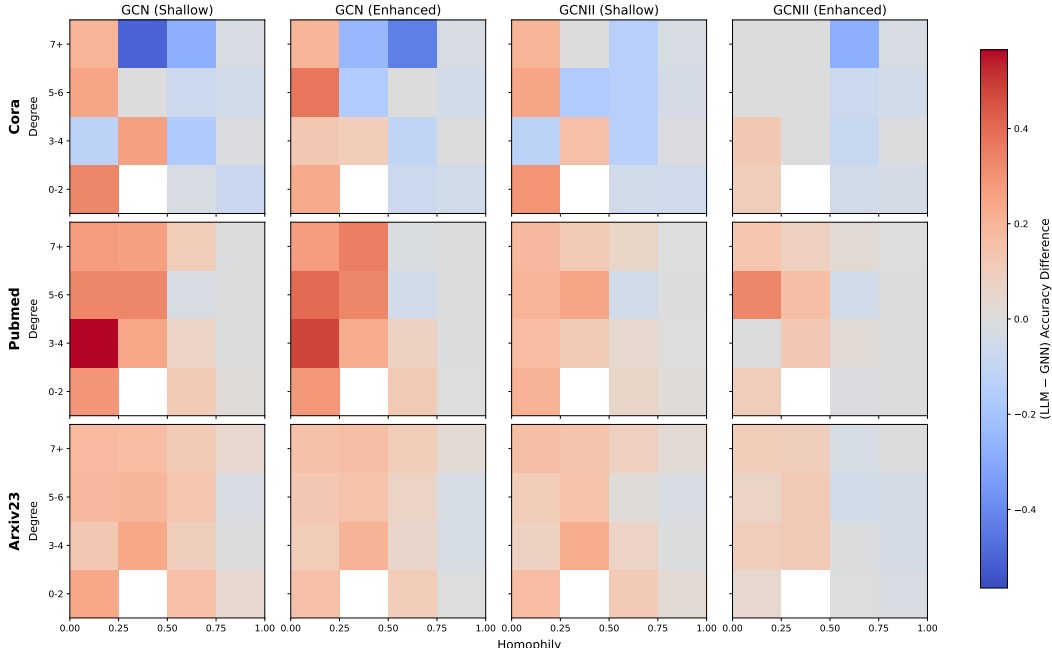

**Figure 4: Stratified Performance Based on Homophily and Degree.** Darker red denotes instances where the LLM performs best, and darker blue denotes instances where the GNN performs best. Across methods, we see further deviation in performance as compared to the individual metrics

### E.3 CORRELATION BETWEEN HOMOPHILY AND UNCERTAINTY

Table 9 shows a moderate correlation (inverse) between model uncertainty and local homophily across all datasets. Intuitively, as homophily decreases (i.e., more label conflict in neighborhoods), uncertainty increases. This empirically links the two most effective routing cues where regions of low homophily tend to be high-uncertainty. However, this correlation is relatively weak, thus we hypothesize including both metrics can provide complimentary information, i.e. the signals are not providing redundant information.

### E.4 STRATIFIED ANALYSIS ON PUBMED

In Figure 5, we observe consistent trends with those reported on Cora and Arxiv23. Specifically, while performance across models converges in the high homophily regime, **LLM models deliver pronounced gains in the low-homophily and low-degree settings**, outperforming strong GNN backbones by substantial margins. For example, LLM-based predictions exceed the next best GNN-enhanced variant by as much as $10.4\%$ on heterophilous nodes ($h_v < 0.20$). These findings further confirm that the advantage of LLM augmentation is not dataset-specific, but generalizes across datasets, highlighting homophily as a reliable signal for routing.

### E.5 HOMOPHILY AND DEGREE INTERPLAY

While homophily and degree display diverging trends between LLM and GNN training, highlighting their complimentary strengths, it also known that their interplay can further inform GNN behavior. In Figure 4, we stratify across both properties for each model and dataset, further demonstrating increased divergence in

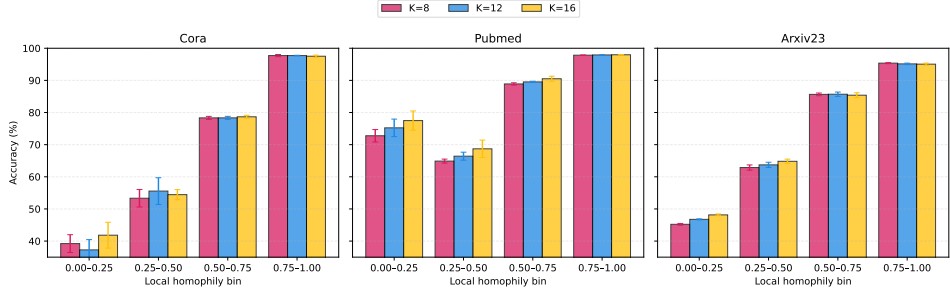

**Figure 6: Stratified Performance for Different K at Test-Time.** We use a batch size of 32 when testing GLANCE, and use different routing budgets depending on $K$. We find that performance tends to increase for heterophilous groups of nodes as we increase $K$, demonstrating GLANCE's ability to take advantage of larger routing budgets.

performance differences. For instance, while Pubmed attains a 10% difference when looking at homophily in isolation, this difference can go as high as ∼30% under the same setting (e.g., for GCNII (enhanced) on Cora, low homophily and low degree can experience a significant improvement as compared to high homophily and high degree).

## F    ADDITIONAL ABLATION AND SENSITIVITY ANALYSES

In this section, we present supplemental analyses introduced in Section 6.3. We begin by examining the sensitivity of GLANCE to the parameter $K$, which controls the number of nodes routed to the LLM at test time. This analysis reveals how performance changes as the budget for LLM queries is adjusted, offering insight into the trade-off between predictive accuracy and computational cost. Understanding this trade-off is particularly important in practice, as practitioners may wish to tune $K$ to align with real-world latency or cost constraints. Then, we ablate the routing features used by GLANCE to determine which signals are most critical for effective routing. By systematically removing each feature, we isolate their individual contributions to routing accuracy and downstream performance. Together, these supplemental analyses deepen our understanding of GLANCE's behavior, confirming both its robustness to different query budgets and its reliance on principled routing features that target GNN failure regimes.

### F.1    PERCENT CHANGE DURING ROUTING

As introduced in Section 6.3, we analyze the effect of varying the routing budget $K \in \{8, 12, 16\}$ on GLANCE's performance. Figure 6 reports stratified results across different levels of local homophily, allowing us to assess how additional LLM queries impact different subpopulations of nodes. We observe that increasing $K$ generally improves performance on heterophilous and low-homophily nodes, where the GNN struggles most. Importantly, with

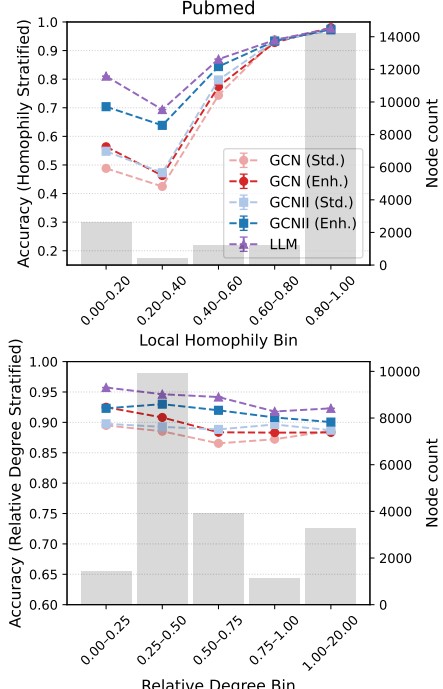

**Figure 5: Stratified Performance.** Performance is given for local homophily (top) and relative degree (bottom); bars denote property distributions (right y-axis).

$K = 16$, GLANCE consistently achieves the highest overall performance across routing strategies, demonstrating that allocating a larger query budget enables the framework to more effectively target the nodes where LLM assistance is most beneficial. This highlights a practical trade-off where larger $K$ increases computational cost but can yield substantial accuracy improvements, particularly in challenging structural regimes.

**Table 10: Stratified and Overall Performance for Standard Routing ($K$=12) and Full Routing ($K$=32).** $\Delta$ denotes the difference from $K$=12 to $K$=32. Green indicates improvement, red indicates a decrease. While full routing can improve heterophilous nodes further, this can come at the cost of performance on homophilous nodes.

| Dataset | | 0.00–0.25 | 0.25–0.50 | 0.50–0.75 | 0.75–1.00 | Overall |
|---|---|---|---|---|---|---|
| Cora | K = 12 | 46.4±2.4 | 54.4±1.6 | 79.7±0.9 | 98.1±0.2 | 89.5±0.4 |
| | K = 32 | 49.2±2.4 | 68.9±3.2 | 76.9±2.1 | 96.9±0.8 | 88.3±1.7 |
| | $\Delta$ | +2.8 | +14.5 | −2.8 | −1.2 | −1.2 |
| PubMed | K = 12 | 71.5±0.5 | 65.9±0.4 | 89.4±0.4 | 97.7±0.1 | 92.6±0.1 |
| | K = 32 | 82.4±0.7 | 79.2±3.3 | 91.1±0.9 | 98.0±0.09 | 94.1±0.2 |
| | $\Delta$ | +10.9 | +13.3 | +1.7 | +0.3 | +1.5 |
| Arxiv23 | K = 12 | 45.2±0.7 | 63.2±0.2 | 85.9±0.3 | 95.3±0.0 | 82.1±0.1 |
| | K = 32 | 49.7±1.84 | 65.8±1.2 | 82.5±1.6 | 94.0±0.2 | 81.6±0.4 |
| | $\Delta$ | +4.5 | +2.6 | −3.4 | −1.3 | −0.5 |

**Table 11: Routed Accuracy Breakdown: GLANCE vs. GCNII.** We use $K = 12$ with a batch size of 32. We compare the accuracy on the subset of nodes routed by GLANCE to the LLM, denoted $\mathcal{R}$, against the baseline where the GCNII backbone makes predictions for the exact same set $\mathcal{R}$. The final row highlights the performance improvement gained by routing these nodes to an LLM via GLANCE. The large positive improvements demonstrate that GLANCE effectively identifies and routes difficult nodes that benefit significantly from LLM inference.

| Model on Routed Set $\mathcal{R}$ | Cora | Pubmed | Arxiv23 |
|---|---|---|---|
| **GLANCE** (Routed to LLM) | 87.6±0.9 | 91.6±0.5 | 77.1±0.8 |
| **GCNII** (Routed to GCNII) | 82.7±1.6 | 85.2±0.9 | 72.6±2.4 |
| **Improvement (GLANCE vs. GCNII)** | +4.9 | +6.4 | +4.5 |

**Full Routing.** To further illustrate the value of selective routing, we evaluate the setting where every node in a batch is routed to the LLM (denoted as $K$=32 for a batch size of 32), allowing the LLM to influence all predictions. As shown in Table 10, full routing is not necessarily beneficial, reinforcing the need for GLANCE's selective strategy. Specifically, while a higher routing budget improves performance on the most challenging nodes (those in the lowest homophily bins) we also observe a decline in accuracy on high homophily nodes. Consequently, both Cora and Arxiv23 see reduced overall accuracy despite the substantially higher query cost. This trend highlights an important trade-off: although LLMs can correct difficult nodes, indiscriminately routing all nodes dilutes accuracy where the GNN already performs well and significantly increases computational cost. In contrast, GLANCE's selective routing (e.g., $K$=12) delivers substantial gains on heterophilous nodes while preserving strong performance on high homophily regions, all with far fewer LLM calls. We also note that while Pubmed does not exhibit as pronounced of a drop, this result is expected as Pubmed's node text has been shown to be highly informative, even enabling LLMs to predict node labels accurately in zero-shot settings (Chen et al., 2024a). However, despite this capability, the performance gap between $K$=12 and $K$=32 remains extremely small in the highest-homophily bin, indicating that the significant increase in computation offers minimal benefit.

## F.2 GRANULAR ANALYSIS ON ROUTING PERFORMANCE

In this section, we provide a deeper analysis on the performance differences between routed and non-routed nodes. Moreover, we offer additional studies to characterize performance under a random router.

### F.2.1 PERFORMANCE IMPROVEMENTS FROM LLM ROUTING

To characterize the effectiveness of GLANCE's routing, we evaluate the predictive accuracy specifically on the subset of nodes routed to the LLM, denoted $\mathcal{R}$. We compare the performance of GLANCE against a baseline where predictions for this exact same set $\mathcal{R}$ are made solely by the GCNII backbone. This direct comparison provides clear evidence for whether invoking the LLM improves prediction accuracy for the nodes selected by our router. The results are shown in Table 11.

**Findings.** We compare the accuracies on the routed nodes $\mathcal{R}$ using predictions from the GCNII backbone versus the LLM. Across all datasets, routing these selected nodes to the LLM yields substantially higher

**Table 12: Comparison of Standard vs. Random Routing for GLANCE.** Random routing selects nodes randomly from each batch for LLM refinement, while Standard uses the router learned within GLANCE. The $\Delta$ accuracy rows report the difference in accuracy for each method across overall, routed, and non-routed accuracy. Positive $\Delta$s (green) indicate improvement with learned routing over random routing. Overall, GLANCE's learned router consistently improves performance, particularly on the non-routed subsets where standard GNNs excel.

| Routing Strategy | Metric | Cora | Pubmed | Arxiv23 |
|---|---|---|---|---|
| **Random** | Overall Acc | 86.4±1.1 | 91.4±0.8 | 80.3±0.9 |
| | Routed $\mathcal{R}$ Accuracy | 83.3±2.3 | 91.9±1.7 | 74.8±0.9 |
| | Non-routed $\mathcal{R}'$ Accuracy | 87.5±0.2 | 91.2±0.3 | 82.1±0.8 |
| **Standard (as used in GLANCE)** | Overall Acc | 89.5±0.4 | 92.6±0.1 | 82.1±0.1 |
| | Routed $\mathcal{R}$ Accuracy | 87.6±0.9 | 91.6±0.5 | 77.1±0.8 |
| | Non-routed $\mathcal{R}'$ Accuracy | 90.2±1.2 | 92.9±0.6 | 84.0±0.7 |
| **$\Delta$ Accuracy (Standard – Random)** | **Overall** | +3.1 | +1.2 | +1.8 |
| | **Routed $\mathcal{R}$** | +4.3 | −0.3 | +2.3 |
| | **Non-routed $\mathcal{R}'$** | +2.7 | +1.7 | +1.9 |

accuracy. Specifically, GLANCE improves performance on $\mathcal{R}$ by +4.9% (Cora), +6.4% (PubMed), and +4.5% (Arxiv23). These significant gains demonstrate that GLANCE's routing policy is highly effective at identifying challenging nodes that directly benefit from the enhanced capabilities of the LLM.

### F.2.2 RANDOM ROUTING PERFORMANCE

To contextualize the routing improvements further, we evaluate a random routing baseline that, for the same query budget ($K=12$), selects nodes at random while keeping all other components of GLANCE unchanged. This experiment serves as a natural lower bound on routing quality and serves as another indicator to whether GLANCE is able to identify difficult nodes. We measure the difference in overall, routed, and non-routed performance between the standard and random routing schemes in Table 12 where more positive is better. We perform 3 runs per experiment and report standard deviations.

**Findings.** We first compare GLANCE's learned router (denoted Standard) against a random routing baseline. Under random routing, overall performance degrades across all datasets, with accuracy dropping by up to 3.1% (Cora). More importantly, accuracy on the routed subset $\mathcal{R}$ is generally lower than with the learned router, suffering drops of 4.3% and 2.3% on Cora and Arxiv23, respectively. This result demonstrates that randomly selected nodes benefit less from LLM refinement. While PubMed exhibits a marginal 0.3% increase on $\mathcal{R}$ under random routing, the non-routed accuracy $\mathcal{R}'$ drops consistently across all datasets. This indicates that a random policy frequently wastes LLM queries on simpler nodes that the base GNN could already classify correctly, leaving behind challenging nodes that the GNN struggles to resolve. Together, these results reinforce that GLANCE's lightweight routing reliably identifies semantically and structurally difficult nodes, yielding targeted performance gains where standard GNNs fail.

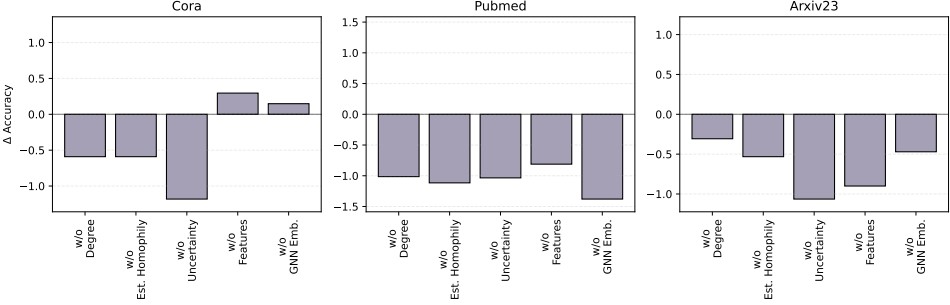

**Figure 7: Ablation Study Over Routing Features - Overall Performance.** Each plot denotes the performance changes, relative to full GLANCE performance, when training without one of the routing features. Performance typically decays across datasets and features, highlighting the benefit of each feature for GLANCE's robust performance.

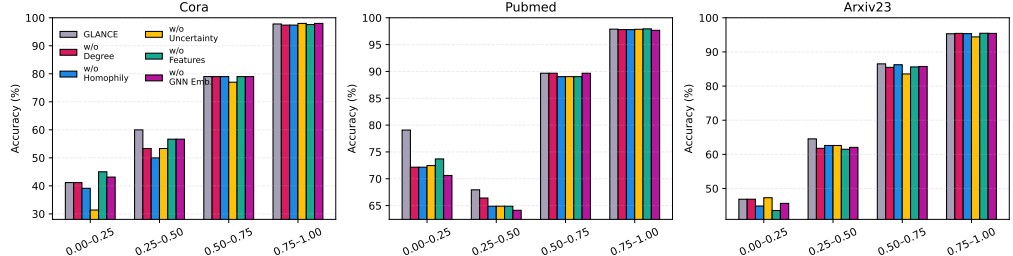

**Figure 8: Ablation Study Over Routing Features - Stratified Performance.** Each set of bars denotes the stratified performance of GLANCE when training without one of the routing features. The gray bar denotes full GLANCE training. We find that the largest performance drops occur in the heterophilous regions, highlighting that GLANCE is specifically targeting these difficult nodes.

**Table 13:** Per-bin accuracy across homophily levels for GLANCE trained with different $\beta$ strength . Larger $\beta$ requires routed nodes to have more substantial impact. We find that increasing $\beta$ tends to systematically increase performance on heterophilous nodes.

| | $\beta$ | Cora | | | | Pubmed | | | | Arxiv23 | | | |
|---|---|---|---|---|---|---|---|---|---|---|---|---|---|
| | | 0.00–0.25 | 0.25–0.50 | 0.50–0.75 | 0.75–1.00 | 0.00–0.25 | 0.25–0.50 | 0.50–0.75 | 0.75–1.00 | 0.00–0.25 | 0.25–0.50 | 0.50–0.75 | 0.75–1.00 |
| | 0.0 | 45.1 | 53.3 | 79.0 | 97.8 | 68.0 | 67.2 | 89.7 | 97.5 | 42.4 | 60.9 | 86.8 | 95.6 |
| GLANCE | 0.1 | 46.4 | 54.4 | 79.7 | 98.1 | 70.1 | 68.7 | 89.2 | 97.6 | 45.2 | 63.2 | 86.0 | 95.3 |
| | 0.2 | 47.1 | 53.3 | 80.0 | 97.8 | 70.6 | 65.7 | 88.6 | 97.7 | 45.2 | 62.9 | 85.2 | 95.5 |
| | 0.3 | 51.0 | 53.3 | 79.0 | 97.8 | 71.3 | 65.7 | 89.2 | 97.8 | 45.5 | 63.2 | 85.1 | 95.6 |

### F.3 ROUTING FEATURE SENSITIVITY

While each routing feature is motivated either by prior work or our analysis in Section 4.1, we quantify their contribution through an ablation study. For each dataset, we retrain GLANCE while removing each feature in turn and measure the resulting change in accuracy. Figure 7 reports the performance drop relative to the full model, showing that overall accuracy consistently declines when any feature is removed. Notably, the impact varies across datasets where no single feature dominates universally, underscoring that different structural signals matter in different settings. This trend is further evident in Figure 8, where, within stratified bins, removing a single feature produces substantial degradations in performance.

### F.4 DIFFERENT ROUTING PENALTIES

In this section, we provide an additional sensitivity analysis on $\beta$, GLANCE's penalty term. Increasing $\beta$, the margin a routed node must surpass for the LLM to be selected, systematically re-allocates LLM usage toward the hardest, most heterophilous nodes. As shown in Table 13, raising $\beta$ improves accuracy in the lowest-homophily bin across Cora (from 45.1 to 51.0), Pubmed (from 68.0 to 71.3), and Arxiv23 (from 42.4 to 45.5), while leaving the most homophilous bin effectively unchanged and producing only small drops in the intermediate bins. In practice, the router spends fewer calls on marginal gains in the intermediary homophily levels where signal is weaker and concentrates budget where the GNN can be most improved upon. This behavior makes $\beta$ a simple, yet effective, knob to trade-off homophilous and heterophilous nodes. When leveraging GLANCE, we recommend tuning $\beta$ on validation metrics in the range (0.1-0.3) to determine the most effective value.

## G SCALING GLANCE TO LARGER DATASETS

Having evaluated GLANCE's performance on small- to medium-scale TAG benchmarks, we now turn to scalability. A concern with LLM–GNN fusion is whether additional components introduced by a method introduce significant overheads, especially when moving to much larger datasets. To address this, we first break down the runtime of GLANCE into its major components: (i) GNN computation, (ii) LLM computation, and (iii) GLANCE-specific modules, namely the router, feature generation, and refinement stages. As shown in Figure 9, the results reveal a clear trend where the LLM dominates the overall runtime,

the GLANCE-specific components are extremely lightweight, constituting only a negligible fraction of total cost. Both routing and refinement involve simple feed-forward operations over low-dimensional features, while feature generation is largely pre-computed within the pipeline. This finding has two key implications. First, it validates our design choice to offload expensive reasoning exclusively to the LLM, while keeping routing and refinement efficient. Second, it demonstrates that scaling GLANCE to larger graphs is primarily limited by the LLM's cost, rather than by the fusion mechanism itself. We therefore use this as motivation to apply GLANCE to the substantially larger dataset OGB-Products seen in Table 5, showcasing its ability to scale while retaining efficiency and accuracy.

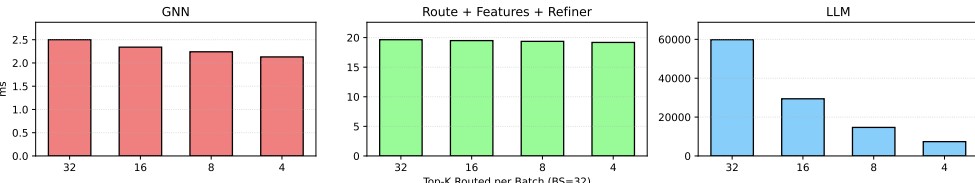

**Figure 9: Runtime breakdown of GLANCE.** LLM computation dominates overall runtime, while GLANCE-specific modules add only negligible overhead, confirming the framework's scalability to larger datasets.

**Computational Cost.** We analyze the computational complexity of GLANCE by isolating the costs of its three key components: (1) the routing features, (2) the router, and (3) the refiner. As GLANCE operates on top of frozen GNN and LLM encoders and is agnostic to their internal architectures, we do not define a specific GNN or LLM cost. Instead, we focus on the GLANCE-specific elements. For each component, we denote the cost of a forward and backward pass, when applicable, by $T^{\text{fwd}}$ and $T^{\text{bwd}}$, respectively.

**Notation.** Let $B$ denote the batch size, $k$ the routing budget per batch, $p$ the dimensionality of the routing feature vector, $c$ the number of classes for a dataset, and $d_{\max}$ the maximum degree of any node in the batch. We also define the following per-node costs:

- $C_{\text{GNN}}$: cost of applying the frozen GNN encoder to one node,
- $C_{\text{LLM}}$: cost of an LLM forward pass for one routed node,
- $C_Q$: cost of applying the homophily MLP to one node,
- $C_{\text{ref}}$: cost of applying the refiner MLP to one routed node.

**Building Routing Features (Per Batch).** Routing features consist of: (1) GNN embeddings, (2) GNN uncertainty, (3) degree information, (4) original node features, and (5) homophily estimates. Using the above per-node costs, this results in the per-batch cost:

$$T_{\text{feat}}^{\text{fwd}} = O\big(BC_{\text{GNN}}\big) + O\big(BRC_{\text{GNN}}\big) + O(B) + O(B) + O\big(B(C_Q + d_{\max}c)\big),$$

where $R$ is the number of stochastic GNN passes used for uncertainty.

**Router.** The router applies a linear transformation to the features and selects the top-$k$ nodes to route:

$$T_{\text{router}}^{\text{fwd}} = O(Bp) + O(B\log B), \qquad T_{\text{router}}^{\text{bwd}} = O(Bp).$$

**Selective LLM Queries.** GLANCE queries the LLM for the $k$ routed nodes (with no backward pass):

$$T_{\text{LLM}}^{\text{fwd}} = O(kC_{\text{LLM}}).$$

**Refiner MLP.** For each routed node, GLANCE applies the refiner MLP:

$$T_{\text{refiner}}^{\text{fwd}} = O(kC_{\text{ref}}), \qquad T_{\text{refiner}}^{\text{bwd}} = O(kC_{\text{ref}}).$$

**Overall Per-Batch Complexity.** Combining all components yields:

$$T_{\text{GLANCE}}^{\text{batch}} = O\Big(\underbrace{B\big((1+R)C_{\text{GNN}} + C_Q + d_{\max}c + p + \log B\big)}_{\text{GNN \& Routing Feature}} + \underbrace{k\big(C_{\text{LLM}} + C_{\text{ref}}\big)}_{\text{LLM and Refiner}}\Big)$$

**Interpretation.** GLANCE's complexity is comprised of two core terms: (1) the full batch processing that creates routing features leveraging the GNN, and (2) the LLM and refiner computation applied only to the $k$ routed nodes. Given $C_{\mathrm{LLM}}$ far exceeds the other terms (as shown in Figure 9), one can scale up the GNN, router, or refiner without incurring large relative increases in runtime. Moreover, because training uses a policy-gradient-style optimization procedure, the cost differential between training and inference is small as the backward pass is only on the cheaper components of the pipeline.

