# OpenReview forum: "Glance for Context: Learning When to Leverage LLMs for Node-Aware GNN-LLM Fusion"
_ICLR.cc/2026/Conference — ICLR 2026 Poster_

### Official Review · Reviewer_r4iU · 2025-10-16

**Soundness:** 3
**Presentation:** 3
**Contribution:** 3
**Rating:** 8
**Confidence:** 3

**Summary:**

This paper studies leveraging LLMs to assist with the node classification task and introduces a novel approach: selectively invoking LLMs for specific nodes. Through an empirical study, the authors identify homophily as a key factor in determining whether predictions should rely on LLMs or GNNs. Building on this insight, the paper proposes the GLANCE framework, which integrates a router component for selecting nodes to be processed by LLMs, a fusion component to combine embeddings from GNNs and LLMs, and a reward-based strategy to optimize both components stably. Extensive experiments on citation networks demonstrate GLANCE's improvements for low-homophily nodes and its scalability to larger e-commerce networks.

This work introduces a promising direction for graph learning, i.e., **selectively invoking LLMs only when necessary** to balance performance and efficiency.

**Strengths:**

1. **This paper is well-motivated and explores a novel research direction**. It is intuitive that GNNs and LLMs are complementary: GNNs excel in high-homophily scenarios, while LLMs can significantly improve predictions for heterophilic graphs (as supported by prior works like [Ref A]). However, no prior work has investigated the selective usage of LLMs to improve performance for heterophilic nodes while maintaining computational efficiency.

2. **The preliminary experiments are comprehensive and provide meaningful insights**. To determine which nodes should be processed by LLMs instead of GNNs, the authors first analyze structural features like degree and clustering density, then identify homophily as the decisive factor. Finally, they propose a label-free metric, estimated local homophily, to guide the routing. This multi-step analysis provides a clear and logical progression for deriving the proposed factor.

3. **The GLANCE framework is carefully designed to align with the problem's motivation and is grounded in rational principles**. During inference, GLANCE achieves efficiency by using only two feature forward passes: one for the router to decide whether to invoke LLMs and another for combining complementary embeddings. During training, the authors incorporate both task-level prediction loss and a policy-gradient-inspired loss to optimize the routing policy, ensuring robust performance.

4. **The experimental evaluation is extensive and promising**, demonstrating GLANCE's improvement in predictive performance, its controllable computational cost, and its scalability to large-scale graphs. The results convincingly show that GLANCE is effective, particularly for low-homophily nodes.

---

[Ref A] Wu, Xixi et al. *When Do LLMs Help With Node Classification? A Comprehensive Analysis*. In ICML, 2025

**Weaknesses:**

1. While the paper is generally well-written and easy to follow, some minor points require clarification. For example:
     * In Figure 1, the notations, e.g., GCN (Std.), GCN (Enh), LLM, are unclear. What do they represent?
     * In Section 5, the authors state that GNNs and LLMs are pre-trained, but the training pipeline is ambiguous. My understanding is that GNNs and LLMs are first trained separately on each dataset using classification loss, and then these two encoders are frozen during the GLANCE framework's joint optimization of the router and feature refiner. If this understanding is incorrect, clarifying the exact training process would be helpful.

2. While the authors provide a general efficiency study to showcase GLANCE’s computational cost, a more detailed analysis would enhance the paper, e.g., what is the computational complexity of GLANCE during inference, and could the authors provide a breakdown of costs for both training and inference?

3. Although the experiments cover datasets of varying scales, they are still limited in diversity. All four datasets exhibit relatively high homophily, meaning heterophilic nodes form only a small portion. As a result, the overall performance improvements from GLANCE are marginal. Extending the experiments to more heterophilic datasets, e.g., the original arXiv dataset, would better validate GLANCE’s effectiveness for heterophilic scenarios.

**Questions:**

1. For Weakness 1, is my understanding of the training pipeline correct? Additionally, could the authors define the notations used in Figure 1?

2. Have the authors considered removing the router loss entirely and jointly optimizing the router and refiner solely based on task prediction loss? If so, how does this alternative approach compare in terms of performance and stability?

---

> ### Author Response · Authors · 2025-11-19
> **Response to r4iU**
>
> We thank you for your positive feedback and are pleased you found our work novel and technically sound. We also appreciate your constructive feedback. Below we address each of your comments.
>
> **W1.** Thank you for the clarifying question. We have clarified Figure 1 by defining all notations directly in the caption and expanding the explanation around line 150 (Section 4.1). Specifically, “Std.” denotes standard GNN training using the shallow textual features typically associated with the datasets (e.g., word2vec-style embeddings), “Enh.” refers to enhanced LLM-based embeddings, and “LLM” corresponds to the LLM-as-Predictor paradigm where node information is serialized as a prompt. Regarding the training pipeline, your understanding is exactly correct. We originally described GNN and LLM pretraining in Appendix C.1 and Appendix C.3 but did not explicitly link them to GLANCE training. We have now made this connection explicit in Appendix C.4 (around line 902).
>
> **W2.** Thank you for pointing this out. In the revised manuscript, we have added a new section in Appendix G that provides a detailed breakdown of GLANCE’s computational cost and distinguishes the components involved in training and inference. At a high-level, we decompose GLANCE’s cost based on three key components: routing feature creation, selective LLM usage, and refiner. We detail the operations that apply to the entire batch (GNN and routing feature operations), differentiating them from those that apply only to the routed nodes (LLM and refiner computation). As the LLM is the most expensive component, this second term dominates the overall computational cost. We additionally clarify that only the router and refiner require backward passes given the policy gradient-style objective, keeping the gap between training and inference small. We also highlight that Fig. 9 corroborates that the LLM calls indeed dominate the cost, while the newly introduced GLANCE modules introduce negligible overhead.
>
> **W3.** Thank you for this suggestion. We have now included a large-scale heterophilous dataset, Arxiv-Year (h = 0.22), coming from the large-scale heterophilous benchmark [1], in Table 5. We have also included expanded discussion around line 466. Notably, this dataset would typically be infeasible for a full LLM-based pipeline. We find that GLANCE improves both heterophilous node accuracy and overall performance with a relatively small query cost (K = 1 with batch size of 64). Together with the prior benchmarks, this new result demonstrates that GLANCE can benefit both homophilous and heterophilous graphs at varying scales.
>
> **Q1.** We believe this has been addressed through our response to W1, please let us know if anything else can be clarified.
>
> **Q2.** We appreciate this question regarding the loss. We first want to point out that the router loss does primarily utilize the task prediction loss, with the core addition being the $\beta$ term acting as a margin to encourage the router to select harder nodes. Intuitively, as $\beta$ increases, GLANCE has to make more substantial improvements to the node’s loss, otherwise it will receive a negative reward. Thus, setting $\beta$ too small can encourage the model to route nodes that only receive marginal loss improvements, wasting computation.
>
> Going beyond this intuitive argument, we have also added a new ablation in Appendix F.4 and Table 13 where we vary $\beta \in \{0, 0.1, 0.2, 0.3\}$ across the datasets in Table 3. We observe that when $\beta = 0$, the performance only slightly improves over the base GNN, indicating that the router is not sufficiently routing nodes that the GNN would have mispredicted. As $\beta$ goes up, we tend to see a smooth transition towards higher performance on heterophilous nodes with similar homophilous node performance, demonstrating that the margin penalty does help prioritize more difficult nodes. We believe this improves the clarity as to why $\beta$ is useful, thanks for the suggestion!
>
>
> Again, thank you for your constructive feedback and suggestions. We hope the additional experiments and clarifications address all the raised points and further substantiate GLANCE’s design. Please let us know if there are any further questions we can answer!
>
> [1] Lim, Derek et al. “Large Scale Learning on Non-Homophilous Graphs: New Benchmarks and Strong Simple Methods” In NeurIPS, 2021.

---

> > ### Comment · Reviewer_r4iU · 2025-11-20
> >
> > Thank you to the authors for the thorough response.
> >
> > I have read both the revised manuscript and the authors' response, and I would like to champion this paper with greater confidence.

---

> > > ### Author Response · Authors · 2025-11-21
> > > **Thank You!**
> > >
> > > Thank you for acknowledging our rebuttal and for your strong support of the work! We greatly appreciate your constructive feedback and suggestions. We believe the paper is both stronger and clearer as a result of your review. Please let us know if any other questions arise.

---

### Official Review · Reviewer_8RQH · 2025-10-31

**Soundness:** 2
**Presentation:** 3
**Contribution:** 2
**Rating:** 4
**Confidence:** 3

**Summary:**

The paper proposes GLANCE, a selective GNN–LLM fusion for text-attributed graphs. A router decides which nodes “deserve” LLM embeddings; others use the base GNN only. The routing signal mixes estimated homophily, uncertainty, degree, and representation cues. A refiner fuses GNN and LLM features when routed. Training uses a cost-aware counterfactual reward that prefers routes which reduce supervised loss under a budget. Experiments on several TAG benchmarks report stronger accuracy on low-homophily regions while keeping LLM calls sparse. Ablations suggest homophily features drive much of the routing benefit, and sensitivity studies show larger budgets help heterophilous nodes most.

**Strengths:**

1) New method.
A simple, modular selective GNN–LLM fusion (router + refiner on frozen backbones) that learns when to invoke LLM embeddings under a budget. Clear cost–accuracy tradeoff and easy to plug into existing TAG pipelines.

2) Extensive experiments.
Broad evaluations across multiple text-attributed graph benchmarks, with routing rates, ablations, and sensitivity to budgets. Results consistently show gains where graphs are hard (low homophily) while keeping LLM calls sparse.

3) Interesting task.
Addresses a timely, practical problem: cost-aware reasoning on text-attributed graphs. Selectively augmenting nodes with LLM features makes TAG learning more applicable to real deployments with tight compute limits.

**Weaknesses:**

**Weakness**

1) **Router vs. eval is entangled.**
You use homophily to route and also to judge gains. That can create circularity. Show results when the router does **not** use homophily (or uses only degree/uncertainty). Also slice by other structures (roles, motifs, k-core, betweenness). The ablation drop for \(h_v<0.5\) strengthens this concern.

2) **Homophily estimator lacks detail.**
You build \(\hat h_v\) with an auxiliary classifier \(Q\). Specify \(Q\)’s architecture and size. Report loss, early stopping, and regularization. Clarify train/val label usage and leakage risk. Add calibration metrics (ECE/Brier).

3) **Missing bounds and fuller ablations.**
Add an **oracle router** (LLM correct, GNN wrong) as an upper bound. Add a **shuffled router** with the same budget as a lower bound. This will contextualize NCS and show headroom. Also run **only-X** and **pairwise** feature ablations for redundancy/synergy. The big drop when removing homophily for \(h_v<0.5\) is not the whole story.


**Suggestions**

1) **Counterfactual reward may be biased.**
It uses labels and batch top-k, and only for routed items. That can skew learning. Consider off-policy estimates, stochastic routing, or periodic global calibration.

2) **Test LLM and prompt diversity.**
You rely on one LLM/embedding and one multi-level prompt. Try other sizes/models and one-level vs. multi-level prompts. This separates routing effects from encoder/prompt artifacts.

3) **Disentangle routing vs. feature gains.**
Add a control that routes “hard” nodes but **does not** call the LLM. Reweight the GNN head only. If gains persist, they may come from attention, not LLM content.

**Questions:**

1) **Batching for top-k?**
How do you form batches—random nodes or topology-aware sampling?

2) **Neighborhood sampling details?**
For ego/1-hop/2-hop text: what sampler, sizes, randomness, seeds, and truncation/token caps?

3) **Refiner capacity and overfitting?**
Since GNN/LLM are frozen, what is the refiner’s depth/width/parameter count vs. the GNN head? Any overfitting on small graphs (e.g., Cora)?

---

> ### Author Response · Authors · 2025-11-19
> **Response to 8RQH - Part 1**
>
> We thank you for your comments and suggestions. We appreciate that you found our method clear and evaluation extensive, and we have clarified key points with supplemental analysis to strengthen the work. Below, we address each concern.
>
> **W1.** We want to clarify that there is no circularity between the routing signal and our evaluation metric. GLANCE uses an **estimated** homophily score derived from pseudo-label similarity, not the ground-truth labels used in evaluation. This ensures the routing decision never accesses evaluation information. Additionally, in Figure 7 of the appendix, we already include the requested ablation where the **estimated** homophily feature is removed (“w/o Est. Homophily”), showing that performance decreases and the estimate is important. Any experiments where we use the true homophily levels, such as those in Table 2, are purely exploratory and not used in our GLANCE model. Regarding “k-core” or “betweenness,” we note that such metrics have already been shown to not be effective [1], which is why we focus on well-studied structural factors that govern GNN performance like homophily and degree [2, 3].
>
> **W2.** We have expanded the description of the classifier Q in Appendix C.1, clarifying that Q is a lightweight MLP (2 layers, 64 hidden units, ReLU activation, dropout = 0.2) trained with cross-entropy loss on the training split, with early stopping based on validation loss to prevent overfitting. Because Q operates per node and is trained only on the training set, there is no risk of label leakage. We further clarify that within GLANCE, Q is used to estimate node similarity via their class probability distribution, not to predict labels directly. As a result, this process remains structure-agnostic and unbiased, as described in Section 4.2.2.
>
> **W3.** Thank you for suggesting additional baselines to better contextualize GLANCE’s routing performance. Following your recommendation, we have now incorporated both an empirical upper and lower bound into the paper. First, we include and evaluate a full-routing setting in which every node is sent through the LLM path in Table 10. Discussion is provided in Appendix F.1. We observe that full routing improves the most heterophilous bins but can harm performance on high homophily nodes and can even hurt overall performance, despite the significantly larger compute cost. This demonstrates that GLANCE already captures most of the available benefits while avoiding unnecessary LLM queries. As a lower bound, we implement the requested random router with the same routing budget. Results are given in Table 12. We find that random routing substantially underperforms GLANCE across routed nodes, non-routed nodes, and overall accuracy, confirming that GLANCE’s routing decisions are meaningfully capturing hard-to-classify regions of the graph that benefit from LLM querying.
>
> In addition to these bounds, we have also expanded our analysis and decomposed routed vs. non-routed performance under GLANCE in Table 11. We find that routed nodes are significantly harder for the GNN alone (up to 11.4% performance drops), and invoking the LLM leads to large improvements on these subsets (up to 6.4% performance gains). This directly quantifies the “headroom” available for routing, as suggested.
>
> Finally, we would like to highlight that we already include feature ablations in Figure 7, which show that removing any one of the features creates a consistent performance degradation. These ablations demonstrate that GLANCE’s routing features are all key to achieving the performance benefits of GLANCE. In other words, if they were redundant, we would expect performance to not drop as one feature can be captured by another.
>
> Together, these additions provide both the upper and lower bounds you asked for and clarify that GLANCE’s learned router is not only effective but already operating at a high-level while remaining significantly more efficient.

---

> > ### Author Response · Authors · 2025-11-19
> > **Response to 8RQH - Part 2 and References**
> >
> > **S1.**  We appreciate this suggestion and note that we use a decaying routing policy, as mentioned in Appendix C.4. Thus, during early stages of training, GLANCE actually sees data points beyond top-k to ensure it isn’t biased to just the top-k subset. For smaller datasets, we route the entire data set during the first epoch, allowing the router and refiner to be sufficiently warmed up. At this point in training, there would be no need for a stochastic router as everything is seen. Moreover, beyond these early training stages, we avoid using stochastic sampling as this would force the LLM to waste computation and parameters learning the “easy” nodes that it will not actually see during inference.
> >
> > **S2.** Thanks for your suggestion. We do not expect appreciable differences to occur due to LLM size, instead it is reasonable to assume that any of the emergent behaviors that come from larger LLMs would also benefit GLANCE, at the cost of querying larger LLMs. Moreover, going any larger than 8B parameters would make it so that one could not train on a single GPU and require more advanced distributed training strategies, hurting applicability.
> >
> > **S3.** Thanks for the suggestion. We point out that this idea of routing hard nodes to a different GNN head is precisely the design of many heterophilous GNNs (e.g. GBK-GNN, which we have as a baseline). As shown in Table 3, these methods consistently underperform GLANCE, confirming that the LLM, not attention reweighting, drives the improvements.
> >
> > To further understand the value of the routing process and LLM, we add an additional sensitivity analysis where we route hard nodes, but simply use the GNN head as opposed to the LLM. This enables us to have a clear measure of performance improvement on the routed nodes via the LLM. In our added Table 11, with accompanying analysis in Appendix F.2.1, we report (i) accuracy on routed nodes when using the LLM, (ii) accuracy on the same routed nodes with the LLM disabled (i.e., GNN-only), and (iii) accuracy on non-routed nodes, allowing us to isolate how predictions improve due to the LLM. We first find that routed nodes are significantly harder than non-routed nodes, with drops of up to 11.4%, when not using the LLM and using a pure GNN-based model. This result demonstrates that the router is correctly identifying challenging nodes and sending them to the LLM. Additionally, when we compare using an LLM vs a GNN on this subset, we are able to attain large performance gains of up to 6.4% on these routed subsets. This again shows the gains come not from the attention, but instead GLANCE.
> >
> > **Q1, Q2, Q3.** We have clarified these details in Appendix C. For neighborhood serialization, we use a random neighbor sampling with a fixed random seed. Additionally, truncation details have been added to Appendix B. We have also clarified that the refiner is a lightweight MLP (2 layers, 64 hidden units, ReLU activation, dropout = 0.2) in C.4, which has significantly fewer parameters than the GNN or LLM and is unlikely to be a source of overfitting.
> >
> > [1] Jaiswal, Ajay et al. “All Against Some: Efficient Integration of Large Language Models for Message Passing in Graph Neural Networks”. https://arxiv.org/pdf/2407.14996, 2024.
> >
> > [2] Yan, Yujun et al. “Two sides of the same coin: Heterophily and oversmoothing in graph convolutional neural networks”. In ICDM 2022
> >
> > [3] Subramonian, Arjun et al. “Theoretical and empirical insights into the origins of degree bias in graph neural networks”. In NeurIPS 2024

---

> > > ### Author Response · Authors · 2025-11-26
> > > **Checking In**
> > >
> > > Hello,
> > >
> > > As we near the end of the discussion period, we wanted to follow up to see whether you had any further questions or comments. In response to the suggestions in your original review, we have added several new experiments and expanded multiple sections. We hope these updates help resolve the ambiguities you noted. Please let us know if there is anything else we can clarify. If our revisions sufficiently address your concerns, we would greatly appreciate your consideration in re-evaluating the work.
> > >
> > > Thank you for your time and consideration.

---

### Official Review · Reviewer_qBh6 · 2025-11-01

**Soundness:** 2
**Presentation:** 3
**Contribution:** 2
**Rating:** 4
**Confidence:** 3

**Summary:**

This paper addresses the inefficiency and uneven performance of uniform GNN-LLM fusion for text-attributed graphs and proposes GLANCE, a node-aware fusion framework. It identifies that GNNs excel at high-local-homophily/high-degree nodes while LLMs outperform on heterophilous/low-degree nodes, with local homophily as a signal for LLM benefit. The proposed approach GLANCE freezes pre-trained GNN and LLM, trains a lightweight router to invoke LLMs on hard nodes selectively, and uses a refiner MLP to fuse embeddings.

**Strengths:**

- The paper proposes a node-aware GNN-LLM fusion mechanism, targeting the inefficiency of uniform LLM invocation and filling a gap in TAG learning.
- The proposed GLANCE takes local homophily as an LLM-benefit signal, combined with a homophily estimation scheme, balancing practical deployment applicability.
- GLANCE achieves good performance across different graph datasets, while freezing pre-trained models ensures scalability for large-scale TAGs like OGB-Products.

**Weaknesses:**

- The label-free homophily estimate relies on a trained MLP to predict node labels. This is rather counterintuitive: since this is essentially a node classification task, why is an MLP capable of fulfilling this role instead of a GNN? Furthermore, additional justification is required regarding how this MLP achieves robust performance. For instance, on low-resource data (with few training labels) or high-noise data (with short/noisy node text). If the MLP’s label predictions are inaccurate, will the estimated homophily become a misleading routing signal?
- Given the complex nature of graphs, uncertainty, local homophily, and degree are not necessarily consistent in their implications for label prediction, as also observed in Appendix E.3. Nodes with high homophily/degree may not exhibit low uncertainty (as well as high prediction success); conversely, nodes with low homophily/degree might show high GNN certainty. The manner in which the router interprets these conflicting signals remains an issue of interpretability.
- To control prompt length, GLANCE samples only 5 neighbors per hop for LLM input. This may result in the discard of critical structural information (e.g., a key neighbor of a heterophilous node that explains its label being excluded from sampling).

**Questions:**

In addition to the questions raised above, I have two more questions to be answered:

- Does the paper employ random sampling rather than structure-aware sampling (such as prioritizing neighbors with high degrees or edge weights)? Could this simplification restrict the LLM’s capacity to capture fine-grained structural correlations?
- Is an adaptive $\beta$ that adjusts based on real-time computational constraints better than a fixed one? Would this affect the tradeoff between heterophilous node accuracy and inference speed?

---

> ### Author Response · Authors · 2025-11-19
> **Response to qBh6 - Part 1**
>
> Thank you for your constructive feedback! We appreciate that you acknowledge the practical benefit of our work and the need to fill the gap in TAG learning. Below we address each of your concerns and questions.
>
> **W1.** Thank you for raising this point. We note that several prior works similarly adopt MLPs, or other graph-agnostic functions, to generate pseudo-labels to estimate homophily [1, 2, 3] as they offer a way to approximate label similarity without structural bias. Additionally, in GLANCE, this MLP is not directly intended to perform node classification. Instead, it provides an estimate of node-level similarity in the class space through our soft-label homophily (the dot product between a node’s predicted class distribution and those of its neighbors), capturing the agreement between nodes regardless of whether the predictions are correct. Using a GNN for this purpose would bias the homophily estimate since GNN message passing tends to amplify similarity among connected nodes, introducing circularity as we would be using a structure-dependent model to infer the property (homophily) we aim to measure. Finally, we emphasize that the estimated homophily is only one of several routing signals, ensuring robustness even if the pseudo-labels are noisy. We make this point clearer on line 236 of the updated paper.
>
> **W2.** We appreciate this observation and would like to clarify that the differing behaviors of homophily, degree, and uncertainty are complementary rather than contradictory. Each captures a different dimension of node difficulty:
> - Homophily reflects how well a node’s neighborhood aligns with its label space.
> - Degree encodes the sparsity of local context available to the GNN.
> - Uncertainty measures the model’s confidence in its current prediction.
>
> Disagreement among these features indicates that they capture non-redundant signals that can jointly inform routing. If they were perfectly aligned, their joint use would offer little benefit given GLANCE’s router learns a weighted combination of these features. Additionally, as shown in our ablation results (Appendix F.2), removing any single feature leads to a measurable drop in performance, confirming that these seemingly “conflicting” signals in fact provide synergy for routing decisions.
>
> **W3 + Q1.** We appreciate this concern and clarify that random neighbor sampling ensures an unbiased, representative view of each node’s local context while avoiding systematic preference for certain node attributes. Moreover, limiting the number of sampled neighbors is essential for controlling LLM prompt length and standard in LLM-as-Predictor models [4, 5]. Conceptually, this also mirrors common GNN neighbor-sampling strategies where a small random subset suffices to approximate the aggregated neighborhood signal.
>
> To further evaluate whether random sampling discards meaningful structural information, we inspected the distribution of node degrees across our datasets during prompt creation. Specifically, as we created prompts, we measured how often we sampled a neighbor that was below our threshold cutoff of 5 neighbors as these would be the only scenarios that would change under a degree-based sampling. For Cora and Arxiv23, only ~15% of neighbors fall below our degree cutoff, and it becomes even smaller for larger datasets like Arxiv-year at ~11%. Thus, we expect that sampling towards high-degree neighbors would have minimal effect in practice as they would be changing very small amounts of the prompt. We further highlight this point by studying the standard deviation of GLANCE in Tables 3 and 4. If random sampling were unstable or omitted essential structural correlations, we would expect GLANCE to exhibit higher variability than the GNN baselines given prompt quality would vary significantly across seeds. Instead, we actually see GLANCE shows lower standard deviations while achieving higher accuracy, indicating sufficient and stable coverage. We also point out that since these graphs are unweighted, edge-weight-based prioritization is not applicable.

---

> > ### Author Response · Authors · 2025-11-19
> > **Response to qBh6 - Part 2 and References**
> >
> > **Q2.** We thank you for this question. We would first like to clarify that $\beta$ is not a mechanism to control inference cost and does not change the number of nodes routed, this is governed by $k$. Instead, $\beta$ guides the router toward more difficult nodes and shapes the loss of the policy. Intuitively, a small $\beta$ makes it easier for the router to attain a positive reward as the difference in prediction loss needed to overcome such $\beta$ is small. However, this can also encourage GLANCE to route nodes that provide marginal loss improvements as it is not penalized for such actions. On the other hand, a larger $\beta$ requires GLANCE to make more substantial improvements that significantly decrease the loss on a node’s prediction, otherwise it would receive a negative reward.
> >
> > To empirically demonstrate this behavior, we have included an additional sensitivity analysis in Appendix F.4 and Table 13 of the paper where we vary $\beta$ between 0 and 0.3 (any larger and we see that nearly all nodes receive negative rewards). We find a systematic improvement in heterophilous nodes (3-5% gains) while generally maintaining homophilous node performance. There are instances at the higher levels of $\beta$ where we hurt homophilous node performance due to overpenalizing the routing, e.g. on Arxiv23, and thus tune $\beta$ in the main experiments to find the appropriate trade-off point.
> >
> > Thank you for your detailed questions and suggestions, we hope that our new experiments and clarifications have addressed your comments. Please let us know if you have any further questions we can address!
> >
> > [1] Zhu, Jiong et al. “Graph Neural Networks with Heterophily” In AAAI, 2021.
> >
> > [2] Du, Lun et al. “GBK-GNN: Gated Bi-Kernel Graph Neural Networks for Modeling Both Homophily and Heterophily” In WWW, 2022.
> >
> > [3] Wu, Yuxia et al. “Exploring the Potential of Large Language Models for Heterophilic Graphs” In NAACL, 2025.
> >
> > [4] Chen, Runjin et al. “Exploring the Potential of Large Language Models for Heterophilic Graphs” In ICML, 2024.
> >
> > [5] Wang, Yuxiang et al. “Exploring Graph Tasks with Pure LLMs: A Comprehensive Benchmark and Investigation” https://arxiv.org/abs/2502.18771, 2025

---

> > > ### Author Response · Authors · 2025-11-26
> > > **Checking In**
> > >
> > > Hello,
> > >
> > > As we approach the end of the discussion period, we wanted to follow up to see whether you had any additional questions or comments. We appreciate the suggestions you shared in your review, they were very helpful in improving the work. In response, we have made numerous changes to clarify the presentation of the paper while also incorporating several new experiments. Please let us know if there is anything further we can clarify. If our updates have sufficiently addressed your concerns, we would be grateful if you would consider re-evaluating the work.
> > >
> > > Thank you for your time and consideration.

---

### Official Review · Reviewer_JFZW · 2025-11-02

**Soundness:** 4
**Presentation:** 4
**Contribution:** 3
**Rating:** 6
**Confidence:** 4

**Summary:**

The paper argues that current GNN-LLM hybrids waste LLM calls by applying a single fusion rule to all nodes, which hides where LLMs actually help. By stratifying nodes with respect to local homophily and relative degree, the authors show that GNNs win in high-homophily/high-degree regions, while LLMs win in heterophilous/low-degree pockets. They then introduce GLANCE learns a lightweight per-node router to route top-k nodes to the LLM and fuses GNN+LLM via a small refiner head. On four datasets, GLANCE shows the improvement and it does so with a small query budget.

**Strengths:**

1. The studied problem of deciding when to invoke an LLM for graph node prediction is timely; the observation that LLM benefits are uneven across nodes is clearly argued and well grounded in the analysis.
2. The proposed GLANCE framework is coherent with routing, context construction, and fusion stages reinforcing one another; learning to gate LLM usage from cheap structure aware signals is a sensible and practical design.
3. The reported gains on heterophilous and low degree subsets, together with consistent improvements in overall accuracy under a limited query budget, indicate that selective LLM supervision actually strengthens node level recognition.

**Weaknesses:**

1. Since GLANCE routes only a subset of nodes to the LLM for cost reasons, it would be useful to report the result which sends every node through the LLM GNN fusion path. Even if this is expensive, such a result could act as an empirical upper bound for the proposed router and show how close the learned routing actually gets to the best case. It would also be helpful to understand the trade-off between budget cost and model performance.
2. Why did the authors include heterophilous GNNs when the datasets in Table 2 are all homophilous, where these models are unlikely to be advantageous? It would be more informative to include genuinely heterophilous or non citation graphs, since the current experiments are mostly on citation networks; this would make the empirical evaluation of the proposed method more convincing.
3. To better understand the quality of the routing decision, it would help to break down accuracy by routed vs non routed nodes. Among the nodes that the router decided to send to the LLM, how many became correct due to LLM input, and how many stayed wrong? Among the nodes that were not routed, how many were already correct under GNN only? This breakdown would show whether the router is actually focusing LLM budget on the hard regions of the graph.

**Questions:**

1. In Section 5.1.1 it is not fully specified how the GNN backbone F is obtained before it is used inside GLANCE. Could the authors explain how it is pre-trained?

I would be happy to raise the score if the above concerns can be addressed.

---

> ### Author Response · Authors · 2025-11-19
> **Response to JFZW**
>
> We thank you for your positive feedback and insightful comments. We are glad to see you found the analysis well-grounded and impactful. Additionally, thank you for your suggestions to improve the analysis. Below we address each of your concerns.
>
> **W1.** Thank you for this suggestion. We have added a full-routing evaluation that sends every node through the LLM fusion path by increasing the routing budget to be equal to the batch size. You can find the full results in Table 10 for K = 12 and K = 32 (batch size of 32), and its accompanying analysis in Appendix F.1. We find that full routing does improve accuracy on the most heterophilous nodes, but this tends to come at the cost of reducing accuracy on high-homophily regions. For Cora and Arxiv23, this actually drops overall performance. These results re-emphasize the core thesis of this work that the practice of routing every node to the LLM does not necessarily provide uniform benefit, and can even be harmful beyond computational costs. Regarding the gap between routing with K = 12 vs 32, we want to highlight that we already show in Figure 6 that this can be partially solved via increasing K, with GLANCE approaching the full routing benefits without incurring the costs in both runtime and performance.
>
> **W2.** Thank you for raising this point. While the datasets themselves are homophilous, they contain many heterophilous nodes, so we included the heterophilous GNNs which have been shown to better generalize to these locally heterophilous nodes [1, 2, 3]. That said, we do agree that including a genuine heterophilous dataset is beneficial to the paper. Thus, we have extended the analysis to include Arxiv-Year (h = 0.22), a dataset from the large-scale heterophilous benchmark [4]. These results have been added to Table 5, as well as the discussion in Section 6.4. We find that GLANCE is still able to achieve better performance on heterophilous nodes as well as overall performance compared to the baselines on Arxiv-Year. Additionally, this dataset is much larger than the datasets typically used in LLM-based graph learning, demonstrating GLANCE’s effectiveness across homophily regimes and scalability.
>
> **W3.** Thank you for this suggestion. To better analyze routing quality, we have added a decomposition of accuracy over routed and non-routed nodes in Table 11 with accompanying analysis in Appendix F.2.1. For each dataset, we report (i) accuracy on routed nodes when using the LLM via GLANCE, (ii) accuracy on the same routed nodes with the LLM disabled (i.e., using GCNII to make prediction), and (iii) accuracy on non-routed nodes, allowing us to isolate how predictions improve due to the LLM. Across the datasets, we first find that routed nodes are significantly harder than non-routed nodes under GCNII (drops of 7.5–11.4% accuracy), demonstrating that the router is sending challenging nodes to the LLM. Importantly, utilizing the LLM provides large performance gains, i.e. 4.5-6.4%, on these routed subsets, confirming that the nodes identified for routing do actually benefit from the LLM.
>
> To further validate that the improvements are not merely due to using an LLM and grounded in the routing process, we also include a random routing baseline in Table 12. In this experiment, we compare standard routing to a random routing procedure. We find random routing consistently underperforms the learned router in overall performance, as well as routed and non-routed performance. These two experiments help demonstrate two core benefits of GLANCE: (1) the router chooses nodes that are typically difficult for a GNN, and (b) the router chooses nodes that can be remedied by the use of an LLM.
>
> We finalize this analysis with a set of experiments highlighting the importance of our $\beta$ parameter to encourage the use of the LLM budget on the hard regions of the graph.  Specifically, we added a new ablation in Appendix F.4 where we vary $\beta \in \{0, 0.1, 0.2, 0.3\}$. We observe that as $\beta$ increases, we tend to see a smooth transition towards higher performance on heterophilous nodes (3-5% gains) with similar homophilous node performance, demonstrating that the margin penalty does help prioritize more difficult nodes.
>
> We hope these three experiments help fully capture the routing behaviors learned by GLANCE and how budget is utilized.
>
> **Q1.** Thank you for pointing out this ambiguity. We originally explained the training of the GNN in Appendix C.1, but did not explicitly link it to the GLANCE training. Ultimately, it is simply trained through the standard supervised loss on the training set. We have expanded this explanation in Appendix C.4 (around line 902) with additional details.
>
> Again, thank you for your thoughtful review and suggestions. We hope the new experimental results and additions to the paper have satisfied all the raised questions. Please let us know if there are any further questions we can answer!

---

> > ### Author Response · Authors · 2025-11-19
> > **References**
> >
> > [1] Du, Lun et al. “GBK-GNN: Gated Bi-Kernel Graph Neural Networks for Modeling Both Homophily and Heterophily” In WWW, 2022.
> >
> > [2] Mao, Haitao et al. “Demystifying Structural Disparity in Graph Neural Networks: Can One Size Fit All?” In NeurIPS, 2023
> >
> > [3] Loveland, Donald et al. “On Performance Discrepancies Across Local Homophily Levels in Graph Neural Networks” In LOG, 2023
> >
> > [4] Lim, Derek et al. “Large Scale Learning on Non-Homophilous Graphs: New Benchmarks and Strong Simple Methods” In NeurIPS, 2021.

---

> > > ### Author Response · Authors · 2025-11-26
> > > **Checking In**
> > >
> > > Hello,
> > >
> > > As the end of the discussion period is coming up, we wanted to check in to see whether you had any additional questions or comments. We appreciate your earlier feedback and have incorporated your suggestions through multiple new experiments and analyses. Please let us know if there is anything further we can clarify. If our updates have sufficiently addressed your concerns, we would be grateful if you could consider re-evaluating the work.
> > >
> > > Thank you for your time and consideration.

---

### Author Response · Authors · 2025-11-19
**Meta Response to All Reviewers and Paper Updates**

We sincerely thank all reviewers for their thoughtful feedback and constructive insights. We are encouraged by the overall positive reception, and we appreciate that all reviewers found our work to be novel, timely, and relevant to the important question of: “How can we more effectively utilize LLMs for text-attributed graph learning?”

We are glad to note the highly positive sentiments towards the extensiveness of our experimentation, with multiple reviewers acknowledging our insightful ablations and sensitivity analyses (JFZW, 8RQH, and r4iU). We are also pleased to see that many reviewers highlight the clarity and logical organization of our work, particularly noting the intuitive flow from our exploratory studies to the design of GLANCE (R4iU, qBh6). Finally, we appreciate the acknowledgement of the practical significance of GLANCE, enabling both scalable and efficient GNN-LLM processing (JFZW, 8RQH, and r4iU).

To address the main questions raised across reviews, we have conducted a series of new experiments and analyses, clarified methodological details, and strengthened our explanations as follows:
- **New Evaluation on a Heterophilous Benchmark (raised by JFZW, r4iU)**. We have added new results to Table 5 and Section 6.4, incorporating the Arxiv-Year dataset from the large-scale heterophilous benchmark. We show that GLANCE provides gains on heterophilous nodes and achieves the best overall performance for Arxiv-Year, complimenting our current results on homophilous benchmarks. Moreover, given Arxiv-Year’s large size, this experiment further demonstrates the scalability of GLANCE.
- **Expanded Sensitivity Study on Router Objective (raised by qBh6, 8RQH, r4iU)**. We have included a new sensitivity analysis in Appendix F.4 and Table 13 where we study the weighting parameter $\beta$ from our advantage objective. Our results show that increasing $\beta$ systematically encourages routing harder (low-homophily) nodes by shaping the reward function, validating the router’s controllability.
- **New Upper and Lower Bounds for Routing (raised by JFZW, 8RQH)**. We provide new ablation studies in Appendix F.2 for (1) full batch routing where all nodes are sent to the LLM, and (2) random routing where nodes are randomly sent to the LLM under the same budget.
    We show that GLANCE achieves superior performance under a learned router with a small budget, emphasizing that selective LLM use is not only more efficient, but also key for performance gains.
- **Clarified Training and Architectural Details (raised by qBh6, 8RQH, r4iU)**.  We provide additional details on the pre-training procedure for the GNN and LLM encoders, as well as the architectures of the homophily estimator and refiner.

The results mentioned above, as well as the individual updates we provide to each reviewer, have been added to the newly updated PDF (new text is in red).

We thank all of the reviewers for their time and effort, please let us know if you have any further questions!

---

### Meta-Review · Area_Chair_JjaE · 2026-01-08

**Summary:**

Most of the reviewers have positive opinion of the paper, recognizing the timeliness of the paper as well as the sound/practical methodology. Experiment results are promisings.

Nevertheless, reviewers noted the following concerns in the original paper:
- Lack of comparison to "full" model that sends every node for LLM fusion.
- Discrepancy between some basslines and datasets (heterophily vs homophily)
- Lack of more detailed results analysis and interpretation.

Overall, the weaknesses are relatively minor and addressable. Authors are encouraged to incorporate their rebuttals into the final camera ready version as far as possible.

**Reviewer Concerns:**

In my opinion, the above concerns are largely addressed by the authors.

**Reviewer Scores:**

probably increase by 0.5-1 on average

---

### Decision · Program_Chairs · 2026-01-26

Accept (Poster)